



# Improved process representation of leaf phenology significantly shifts climate sensitivity of ecosystem carbon balance

Alexander J. Norton[1], A. Anthony Bloom[1], Nicholas C. Parazoo[1], Paul A. Levine[1], Shuang Ma[1], Renato K. Braghiere[2,1], and Luke T. Smallman[3]

[1]Jet Propulsion Laboratory, California Institute of Technology, Pasadena, CA 91101, USA
[2]Division of Geological and Planetary Sciences, California Institute of Technology, Pasadena 91125, USA
[3]School of GeoSciences and NCEO, University of Edinburgh, Edinburgh, UK

**Correspondence:** Alexander J. Norton (alexander.j.norton@jpl.nasa.gov), A. Anthony Bloom (alexis.a.bloom.jpl.nasa.gov)

**Abstract.**

Terrestrial carbon cycle models are routinely used to determine the response of the land carbon sink under expected future climate change, yet these predictions remain highly uncertain. Increasing the realism of processes in these models may help with predictive skill, but any such addition should be confronted with observations and evaluated in the context of the aggregate

behavior of the carbon cycle. Here, two formulations for leaf area index (LAI) phenology are coupled to the same terrestrial biosphere model, one is climate agnostic and the other incorporates direct environmental controls on both timing and growth. Each model is calibrated simultaneously to observations of LAI, net ecosystem exchange (NEE), and biomass using the CARbon DAta-MOdel fraMework (CARDAMOM), and validated against withheld data including eddy covariance estimates of gross primary productivity (GPP) and ecosystem respiration ($R_e$), across six ecosystems from the tropics to high-latitudes.

Both model formulations show similar predictive skill for LAI and NEE. However, with the addition of direct environmental controls on LAI, the integrated model explains 22% more variability in GPP and $R_e$, and reduces biases in these fluxes by 58% and 77%, respectively, while also predicting more realistic annual litterfall rates, due to changes in carbon allocation and turnover. We extend this analysis to evaluate the inferred climate sensitivity of LAI and NEE with the new model, and show that the added complexity shifts the sign, magnitude, and seasonality of NEE sensitivity to precipitation and temperature. This

highlights the benefit of process complexity when inferring underlying processes from Earth observations and in representing the climate response of the terrestrial carbon cycle.

## 1 Introduction

Terrestrial ecosystems play a critical role in the Earth's climate system due to their varied couplings to and feedbacks between carbon, water, and energy with the atmosphere. Improving our ability to quantify and predict the response of terrestrial ecosys-

tems to climate is essential to advancing our understanding of these feedbacks and predicting future climate change (Booth et al., 2012). Despite their importance, there remains considerable uncertainty in our understanding of the terrestrial carbon cycle, undermining our ability to make accurate predictions of future carbon-climate feedbacks (Friedlingstein et al., 2014; Huntzinger et al., 2017; Piao et al., 2013).



Vegetation phenology is a key component of terrestrial ecosystem dynamics as it is directly linked to key processes in the

carbon, water, and energy cycles (e.g. photosynthesis, autotrophic respiration, evapotranspiration, and surface albedo), making it an area of focus in understanding the climate response of ecosystems. Phenology refers to the timing of periodic events in plant development such as reproduction, bud burst, canopy senescence, activity-dormancy cycles, and carbon allocation. Quantitatively, the leaf area index (LAI) represents the one-sided surface area of leaves per area of ground surface. LAI is a cornerstone biophysical quantity for monitoring vegetation phenology, as it can be observed globally from space, and for

representing the canopy in terrestrial biosphere models (TBMs), a key component of Earth system models (Sellers et al., 1997). LAI mediates the canopy interception of radiation, thus it directly controls processes such surface albedo and the rates of photosynthesis and transpiration. Indirectly, LAI also has significant impacts such as influencing how much precipitation reaches the soil surface altering plant available water and evaporation from the soil and canopy surfaces. It also represents the mass of foliar carbon in the canopy, coupling these processes to the cycling of carbon within the plant and litterfall that supplies

carbon to the soil (Richardson et al., 2013).

A variety of concepts have been used to represent LAI dynamics in TBMs, including ecohydrological equilibrium (Yang et al., 2018), optimality principles such as the maximization of plant net carbon gain (Caldararu et al., 2014; Manzoni et al., 2015), direct carbon-supply (Xin et al., 2020), demand for growth (Schiestl-Aalto et al., 2015), and approaches that consider climate and biophysical controls more directly (Jolly et al., 2005; Knorr et al., 2010; Stöckli et al., 2008). Mechanistic modeling

approaches are lacking, a reflection of our limited fundamental understanding of processes such as bud burst, leaf longevity, and canopy senescence and their variability across species (Cooke et al., 2012). Observations have shown that the dynamics of LAI are correlated with environmental variables (Clelend et al., 2007); in particular temperature, water availability, and photoperiod (Delpierre et al., 2016; Iio et al., 2014; Richardson et al., 2013), although variability also occurs within and across species for a given climate (Cole and Sheldon, 2017; Marchand et al., 2020). In the absence of mechanistic understanding,

it is important that model formulations are generalized and calibrated to available observations. Therefore, many TBMs use semi-empirical representations of LAI which depend upon an understanding of these correlations. While many of these models have shown fidelity in representing LAI dynamics (e.g. Stöckli et al., 2008), vegetation phenology remains a large source of uncertainty in models and is therefore an ongoing area of research (Migliavacca et al., 2012; Richardson et al., 2012; Seiler et al., 2022).

In the context of carbon-climate feedbacks, it is critical to understand the role of LAI phenology in mediating the carbon cycle, particularly net ecosystem exchange of carbon (NEE) (Richardson et al., 2012). However, few studies have investigated whether a more complex model representation of LAI can actually improve predictions of NEE or how these improvements affect the sensitivity of the terrestrial carbon balance to climate. These additional steps are needed to evaluate how the representation of specific processes in models ultimately affect the integrated response of NEE to climate (Fisher and Koven, 2020).

Neglecting these steps runs the risk of biased predictions of future carbon-climate feedbacks.

In this study we use a Bayesian data assimilation system to generate a data-informed representation of LAI, its coupling to NEE, and the climate sensitivity of both LAI and NEE. Data assimilation or model-data fusion (MDF) provides a framework for systematically combining observations with a model (Rayner et al., 2019). For understanding a complex system like the ter-





restrial carbon cycle, MDF is a useful approach to improve model performance and mechanistic understanding by constraining
the diverse set of processes, and their interactions, contributing to carbon exchange (Fisher and Koven, 2020; MacBean et al.,
2016).

For the present study, we consider two key aspects of model uncertainty: (i) that the model formulation must accurately repre-
sent the main processes that govern LAI, NEE, and their response to climate (Schwalm et al., 2019), and (ii) that the parameters
of the model must be appropriately assigned (Prentice et al., 2015). For a given model, MDF can provide a parameterization
that is statistically consistent with observations and their uncertainties. When applied equally across multiple model structures,
MDF can be used to evaluate different model structures and their impact on the data-informed processes (e.g. Famiglietti et al.,
2021). Here, we investigate two model formulations for LAI and perform MDF at six flux tower sites distributed across diverse
ecosystems from the tropics to high-latitudes (Baldocchi, 2008). We implement a prognostic, climate-sensitive LAI model into
a TBM and benchmark this against an empirical diagnostic LAI model used in a previous version of the same TBM (Bloom
and Williams, 2015; Quetin et al., 2020). We constrain both models using multiple observations of carbon states and fluxes,
and then use these data-informed models to infer the climate sensitivity of NEE. The main objectives of this study are to: (i)
Investigate the impact of a more complex process representation of LAI on predicting LAI and NEE dynamics in a MDF sys-
tem, (ii) evaluate the impact of a more complex process representation of LAI on inferring the processes underlying NEE: GPP
and $R_e$, and (iii) evaluate how a change in LAI process representation affects the climate sensitivity of the terrestrial carbon
cycle at seasonal and annual time-scales.

## 2   Methods

### 2.1   Study Sites

The study is focused at six sites distributed from the tropics to high-latitudes (Fig. 1) that are part of a global network of
eddy covariance flux sites, FLUXNET (Pastorello et al., 2020). These sites cover a range of climate zones and phenological
strategies (Table 1), allowing for more robust model evaluation and climate sensitivity analysis in the global context. Following
Famiglietti et al. (2021), we selected these sites based on the following criteria: (i) meteorological forcing data availability,
(ii) observational data availability including repeat woody biomass observations and eddy covariance measurements of car-
bon dioxide and water vapor fluxes, (iii) temporal coverage of at least ten years, (iv) no intensive human management (e.g.,
agriculture or logging), and (v) vegetation is dominated by C3 photosynthetic pathway.

### 2.2   Model-data fusion

To quantitatively evaluate the impacts of the process representation of LAI on the net carbon balance and its climate sen-
sitivity, we utilized the CARbon DAta-MOdel fraMework (CARDAMOM, Bloom and Williams, 2015; Bloom et al., 2016).
CARDAMOM is a Bayesian MDF system, used to retrieve time-invariant parameters and initial conditions for the Data Assim-
ilation Linked ECosystem model (DALEC) and has been used widely as a diagnostic tool to infer stocks and fluxes of carbon



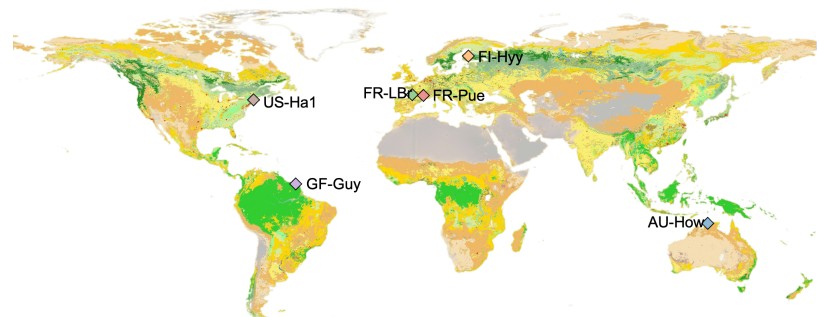

**Figure 1.** Study site locations overlaid onto the land cover type map from the Moderate Resolution Imaging Spectroradiometer, MODIS, 2016 (MCD12C1, Friedl and Sulla-Menashe, 2015).

**Table 1.** Description of the study sites including the site name, FLUXNET code, dominant plant functional type (PFT), location, mean annual temperature (MAT) and mean annual total precipitation (MAP).

| Site | Id | Dominant PFT | Lat, Lon (°) | MAT (°C) | MAP (mm) | Study Period |
|------|-----|--------------|--------------|----------|----------|--------------|
| Howard Springs | AU-How | Tropical woody savanna | -12.49, 131.15 | 27.0 | 1449 | 2001-2014 |
| Hyytiala | FI-Hyy | Boreal evergreen needleleaf | 61.85, 24.29 | 3.8 | 709 | 1999-2014 |
| Le Bray | FR-LBr | Temperate evergreen needleleaf | 44.72, -0.77 | 13.6 | 900 | 1998-2008 |
| Puechabon | FR-Pue | Temperate evergreen broadleaf | 43.74, 3.60 | 13.5 | 883 | 2000-2014 |
| Guyaflux | GF-Guy | Tropical evergreen broadleaf | 5.28, -52.92 | 25.7 | 3041 | 2004-2018 |
| Harvard Forest | US-Ha1 | Temperate deciduous broadleaf | 42.54, -72.17 | 6.2 | 1071 | 1998-2012 |

and water (Bloom et al., 2020; Quetin et al., 2020; Yin et al., 2020; Smallman et al., 2021). CARDAMOM has the capability to assimilate a diverse range of observations (Bloom et al., 2020; Famiglietti et al., 2021) and shows comparable performance to more complex terrestrial biosphere models when it is constrained by observations (Quetin et al., 2020). It has advantages over other MDF frameworks as it does not rely on definitions of plant functional types or on steady-state assumptions.

CARDAMOM is also capable of utilizing different DALEC model formulations (Famiglietti et al., 2021). A number of
DALEC model versions have been developed for various purposes (Williams et al., 2005; Famiglietti et al., 2021). Recent developments have incorporated more processes, as CARDAMOM is increasingly being used to diagnose climate sensitivity of terrestrial carbon and water cycles (Bloom et al., 2020; Ge et al., 2022; Smallman and Williams, 2019; Yang et al., 2022). With this in mind, we implemented a climate-sensitive LAI phenology model into a version of DALEC, which is described further below.





## 2.3   Observations and Model Forcing

Multiple types of observations were used to constrain the processes relevant to LAI, NEE, and their interactions. This helps prevent over-fitting and ensures a consistent view of the terrestrial carbon cycle is achieved between model and data (Kaminski et al., 2013). Observations used include monthly LAI, monthly NEE, and annual woody biomass. Observations of LAI were retrieved from the Earth Observation Copernicus 1 km gridded product over each site, which includes a time-varying uncertainty estimate that we utilized (Fuster et al., 2020; Verger et al., 2014). Observations of NEE using the eddy covariance technique are from the FLUXNET2015 database (Pastorello et al., 2020). The time-varying uncertainty estimate for NEE was based upon propagating instrumentation error (0.58 g C m$^{-2}$ d$^{-1}$ Hill et al., 2012) and temporal aggregation due to missing sub-monthly time steps. Uncertainty due to temporal aggregation was estimated based on site specific statistical models derived from sub-sampling time periods without missing values. Aggregation and instrumentation uncertainty was then combined assuming uncertainties are fully correlated. The in-situ woody biomass observations were converted into estimates of above- and below-ground biomass (ABGB) using allometric scaling based on principle investigator advice at each site. Further details on all of the observations used can be found in Famiglietti et al. (2021).

The study period at each site ranges from 11 to 16 years (Table 1). The data was split into two periods, a training window (calibration) and a prediction window (validation). The first five years was used for calibration and the remaining data was used for validation. Climate forcing data for the model consisted of downward shortwave radiation, air temperature (average 2 m minimum and maximum), precipitation, vapor pressure deficit, and atmospheric carbon dioxide concentration.

To support model evaluation, we used gross primary productivity (GPP) and ecosystem respiration (R$_e$) fluxes from the FLUXNET partitioned NEE (Pastorello et al., 2020). We selected GPP and R$_e$ estimates derived using night-time partitioning, i.e. R$_e$ is determined as a night time R$_e$ fitted to a function of temperature extrapolated into the day time, thus, GPP is estimated as the residual between R$_e$ and NEE. This data was withheld from the model calibration step, thus providing a stringent metric of model skill in representing the processes governing carbon cycling. There are three primary motivations for withholding this data for validation: (i) These are model-based products, (ii) the NEE observations and the GPP and R$_e$ estimates are not wholly independent, and (ii) we only assimilate observations that can be produced directly from Earth observations to permit global application of this framework in future work.

## 2.4   Model Description

The DALEC model version used here has been fully described elsewhere (Bloom and Williams, 2015; Bloom et al., 2016; Quetin et al., 2020; Yang et al., 2022; Yin et al., 2020). We describe, in brief, the representation of the carbon cycle in DALEC. Full description of the water balance can be found in Bloom et al. (2020), which includes a plant-available and a plant-unavailabale soil water pool. Following this, we describe the two separate implementations of LAI phenology used in this study that are linked to same representation of carbon and water cycles. Common model parameters between the two models are shown in Table A1.




### 2.4.1 Carbon Balance

The carbon cycle in DALEC consists of six carbon pools (labile, foliar, wood, fine root, litter, and soil) and simulates pool transfers using ordinary differential equations. The NEE of an undisturbed ecosystem is calculated as:

135
$$NEE = (R_h + R_a) - GPP = R_e - GPP \tag{1}$$

Where GPP is the gross primary productivity, $R_h$ is the heterotrophic respiration from litter and soil carbon, $R_a$ is the autotrophic respiration, and $R_e$ is the ecosystem respiration representing the sum of $R_h$ and $R_a$. The representation of GPP is based on the Aggregated Canopy Model (Williams et al., 1997, ACM), with the specific implementation described in (Bloom et al., 2016). The ACM model is a parsimonious approach for representing GPP with calibration (Williams et al., 1997). It 140 requires inputs of temperature, carbon dioxide concentration, downward shortwave radiation, and LAI. The GPP simulated by ACM is then scaled by a soil moisture limitation factor that is calculated using the plant available soil water and a parameter for the wilting point.

### 2.4.2 LAI Phenology Models

The focus of this study is upon the representation of LAI phenology. We implemented a new model for LAI phenology into 145 DALEC, based upon the model of Knorr et al. (2010). As a benchmark, we also utilized a diagnostic LAI phenology model commonly used in CARDAMOM studies, the Combined Deciduous-Evergreen Analytical model (CDEA). These two DALEC model formulations are denoted DALEC$_{\text{Knorr}}$ and DALEC$_{\text{CDEA}}$, respectively.

**CDEA Model**

The CDEA model is a relatively simple model for simulating the phenology, growth, and turnover of LAI, with full descriptions 150 described elsewhere (Bloom and Williams, 2015; Famiglietti et al., 2021; Quetin et al., 2020), with the specific formulation the same as that of Bloom et al. (2020). In brief, the CDEA model computes leaf onset and leaf fall factors that govern the flux of carbon from the plant labile carbon pool ($C_{\text{lab}}$) to the foliar carbon pool ($C_{\text{fol}}$) and flux of carbon from $C_{\text{fol}}$ to the litter carbon pool ($C_{\text{lit}}$), respectively. Carbon can also be supplied directly from GPP via a fractional allocation parameter. The leaf onset and leaf fall factors are based on a day-of-year approach that govern the timing of phenological events, including peak day of 155 year for labile turnover (supporting leaf growth) and for foliar turnover (controlling litterfall). A parameter that governs the leaf longevity determines how much of the canopy is turned over each year. The CDEA model is a relatively simple and generic representation of LAI phenology and consists of eight parameters as well as two initial condition parameters for $C_{\text{lab}}$ and $C_{\text{fol}}$ (Table B1). There is no representation of direct environmental control on the timing of phenological events (e.g. spring onset, fall senescence), hence these are fixed from year to year. However, environmental effects on GPP can propagate through to 160 LAI via changes in carbon allocation, thus allowing for the magnitude of LAI to be indirectly sensitive to climate via changes in carbon supply.



**Knorr Model**

The new LAI phenology model implemented into CARDAMOM is a development of the model by Knorr et al. (2010). This is a prognostic model governed by environmental constraints on the timing and growth of the canopy. Full description of the model can be found in Knorr et al. (2010). Here, we briefly describe the model and its novel implementation in DALEC which includes coupling to the carbon and water cycles. The Knorr LAI model as implemented in DALEC consists of ten parameters as well as four initial condition parameters (Table C1).

The prevailing understanding of the dynamics of LAI across global biomes is that there are three primary environmental controls: temperature, photoperiod, and water availability (Richardson et al., 2012). The Knorr model considers all three as potentially limiting factors, with the specific dynamics governed by climate and model parameterization. In addition, we couple LAI phenology to the plant carbon balance and incorporate a function for carbon supply limitation on LAI growth thus providing a fourth potentially limiting factor.

**Representing Activity-Dormancy Triggers in a Population**

A common approach to modeling leaf phenology is to use one or more growth triggers (or thresholds) that transition a plant into or out of an active growth state. This is problematic as it is often modeled using a discrete, binary formulation, which makes these functions unrealistic when representing a population of individuals (e.g. within a model grid cell). In reality, plants within a given population do not reach these thresholds simultaneously (Cooke et al., 2012), thus a distribution of threshold values to represent the population of individuals is more realistic, and this is likely to result in a relatively smooth transition toward the new growth state when integrated over the population. A discrete formulation is also non-differentiable, which is problematic for derivative-based MDF techniques. Knorr et al. (2010) developed a convenient solution to this problem by representing threshold parameters with a normal distribution in space.

Two temperature thresholds, one for temperature and one for photoperiod, are each represented by a cumulative normal distribution function ($\Phi$). The multiplication of these two cumulative normal distributions gives the fraction of individuals within the population that are in an active growth state, f, as:

$$f = \Phi\left(\frac{T - T_\phi}{T_r}\right)\Phi\left(\frac{t_d - t_c}{t_r}\right) \tag{2}$$

Where $T$ represents the air temperature memory, analogous to the growing degree days concept (Eq. 20, Knorr et al., 2010), $T_\phi$ is a parameter representing the mean temperature threshold for leaf onset, $T_r$ is a parameter representing the one sigma spatial range of $T_\phi$, $t_d$ is the day length, $t_c$ is a parameter representing the mean day length threshold for leaf senescence, $t_r$ is a parameter representing the one sigma spatial range of $t_c$.

**Temporal Dynamics of LAI**

The temporal evolution of LAI is represented by the following ordinary differential equation:

$$\frac{dLAI(t)}{dt} = \xi\left(LAI_{max} - LAI(t)\right)f - \frac{LAI(t)}{\tau_L}(1-f) \tag{3}$$





where $\tau_L$ is a parameter describing the longevity of leaves during senescence and $\mathrm{LAI_{max}}$ is the maximum potential LAI computed as:

$$LAI_{max}(t) = \nu\big(\hat{\Lambda}, LAI_W\big) \tag{4}$$

Where $\hat{\Lambda}$ is a parameter describing the maximum LAI, as limited by factors such as structure, and $LAI_W$ is the LAI based upon water availability, computed by:

$$LAI_W(t) = \frac{W LAI(t)}{E \tau_W} \tag{5}$$

Where $W$ represents the plant available soil water, $E$ is the evapotranspiration rate, and $\tau_W$ is a parameter representing the expected length of water deficit periods tolerated before leaf shedding. We note that this differs from the original formulation for water limitation such that we use $E$ instead of transpiration as in Knorr et al. (2010), considering this version of DALEC does not differentiate the two.

**Coupling to the Carbon Balance**

While the fundamentals of the Knorr model are grounded in biophysical concepts for activity-dormancy of individuals in a population of plants, it only predicts the net change in LAI. Coupling LAI dynamics to the carbon cycle requires additional assumptions which were not defined in the original model description (Knorr et al., 2010). First, in both DALEC$_{\mathrm{CDEA}}$ and DALEC$_{\mathrm{Knorr}}$, LAI is related to C$_{\mathrm{fol}}$ via a parameter for the leaf mass of carbon per area, LMA, as follows:

$$C_{fol} = LAI \times LMA \tag{6}$$

Second, we must consider that inputs to C$_{\mathrm{fol}}$ come from plant carbon allocation and outputs go to C$_{\mathrm{lit}}$, with the rate of change in C$_{\mathrm{fol}}$ represented by:

$$\frac{dC_{fol}(t)}{dt} = F_{C,lab2fol} - F_{C,fol2lit} \tag{7}$$

Where F$_{\mathrm{C,lab2fol}}$ represents the flux of carbon from C$_{\mathrm{lab}}$ to C$_{\mathrm{fol}}$, and F$_{\mathrm{C,fol2lit}}$ represents the flux of carbon from C$_{\mathrm{fol}}$ to C$_{\mathrm{lit}}$.

The carbon supplied via net primary productivity (i.e. GPP - R$_{\mathrm{a}}$) into C$_{\mathrm{lab}}$ provides the substrate to grow new leaves. To represent F$_{\mathrm{C,lab2fol}}$ we consider both the supply and demand of carbon for new foliar growth. The supply of labile carbon is the sum of new labile production at time $t$ (Eq. C1) and C$_{\mathrm{lab}}$ at end of the previous time step (expressed as a flux over the time step, $\Delta t$), represented by:

$$F_{C,fol,supply}(t) = F_{labprod}(t) + \frac{C_{lab}}{\Delta t} \tag{8}$$





This formulation implies that the entire $C_{lab}$ pool is available for foliar growth at any given time step which is consistent
with findings that $C_{lab}$ does not follow first-order decay kinetics (Martínez-Vilalta et al., 2016). We do not consider constraints
on the release of stored labile carbon such as the phloem loading rate (e.g. Trugman et al., 2018).

For the demand for new foliar growth, we make the assumption that there is a gross demand flux of carbon from $C_{lab}$ to $C_{fol}$
when the canopy LAI is in a net growth state. Conversely, when the canopy (LAI) is in a net senescent state, there is zero gross
demand flux of carbon from $C_{lab}$. This is represented by:

$$F_{C,fol,demand}(t) = max(0, LMA\frac{dLAI(t)}{dt}) \tag{9}$$

Where $\frac{dLAI(t)}{dt}$ is computed from Eq. 3. The actual $F_{C,lab2fol}$ is computed as the smoothed minimum of the supply and
demand fluxes as follows:

$$F_{C,lab2fol}(t) = \upsilon\big(F_{C,fol,supply}(t), F_{C,fol,demand}(t)\big) \tag{10}$$

Where $\upsilon$ represents a quadratically smoothed minimum function (see Eq. C2). This formulation ensures that new foliar
growth only occurs when carbon substrate is available.

The flux $F_{C,fol2lit}$, or litterfall flux, also depends on whether the canopy is in a phase of growth or senescence. Here, we
incorporate an additional term that is necessary to represent litterfall. We note that when the Knorr model is in a fully active
growth phase (i.e. $f = 1$,) which may occur during canopy closure or in evergreen systems, the model (Eq. 3) would predict zero
LAI loss and hence zero litterfall. Observations across the major global biome types show that litterfall never goes completely to
zero (Zhang et al., 2014), as leaves are being turned over constantly at a rate governed by factors such as longevity, herbivory,
and disturbance, even if its not evident from ecosystem scale LAI observations (see Albert et al., 2019). To overcome this
limitation, we add an additional term for loss of LAI via a nominal background turnover rate ($\theta_{fol}$). Therefore, the litterfall
flux is computed as:

$$F_{C,fol2lit}(t) = \begin{cases} \theta_{fol}C_{fol}, & \text{if } \frac{dLAI(t)}{dt} > 0. \\ LMA\frac{dLAI(t)}{dt} + \theta_{fol}C_{fol}, & \text{otherwise.} \end{cases} \tag{11}$$

This formulation ensures that some litter production occurs regardless of the growth-senescence state of the Knorr LAI
model while ensuring conservation of mass.



### 2.4.3   Optimization Algorithm

Following previous CARDAMOM efforts, we jointly retrieve the probability distribution of DALEC time-invariant process parameters and initial state conditions (henceforth vector $x$) given the observational constraints (henceforth vector $O$) using a standard Bayesian inference formulation, where;

$$p(x|O) \ p(x)p(O|x); \tag{12}$$

$p(x)$ is the prior probability distribution of $x$, and $p(O|x)$ is proportional to the likelihood of $x$ given observations $O$ ($L(x|O)$). The prior probability of $x$, $p(x)$ is characterized as the product of (i) a log-uniform prior distribution based on ecologically plausible minimum and maximum values, and (ii) ecological and dynamical constraints (EDCs), where $p(x)$ is equal to 0 if DALEC parameter combinations or simulation outputs meet ecological conditions; these are described in (Bloom and Williams, 2015) and (Bloom et al., 2016).

For a given DALEC run, the likelihood, $L(x|O)$, is defined as:

$$L(x|O) = L_{LAI}L_{NEE} \tag{13}$$

Where $L_{\text{LAI}}$ and $L_{\text{NEE}}$ are the model-observation mismatches. Each likelihood term is derived as:

$$L* = exp(-\frac{1}{2} * \sum ((M_i - O_i)^2/U_i^2)) \tag{14}$$

Where $M_i$, $O_i$ and $U_i$ represent the model output, corresponding observation, and uncertainty, which represents the combined effects of model and observation error on model-data mismatch.

To sample $p(x|O)$, we used a Differential Evolution metropolis hastings Markov Chain Monte Carlo (DE-MCMC) algorithm (Ter Braak, 2006) with 200 walkers (Levine et al., in prep); in previous efforts we used a adaptive metropolis hastings MCMC (Haario et al., 2001). We found that the two algorithms overall give statistically similar results with comparable run times, however the DE-MCMC algorithm was found to be more stable and less likely to generate chains trapped in local minima.

### 2.5   Model Analysis and Diagnostics

### 2.5.1   Parameter Uncertainty Reduction

Following model calibration using CARDAMOM, it is useful to evaluate the constraint that the observations provide upon the model parameters. The prior probability density function (PDF) for the parameters is log uniform between the assumed minimum and maximum prior limits. The posterior PDFs for each parameter are represented by the sub-sampled solutions from the CARDAMOM optimization, hence the posterior PDF can take any form. Uncertainty reduction of the parameters was quantified using the relative change in interquartile range (IQR) from the prior to posterior, given by $100 \times (1 - \frac{IQR_{post}}{IQR_{prior}})$.



Note that this is calculated after transforming posterior parameter sub-samples into log space, to ensure consistency with the
prior PDFs. The relative change metric is analogous to the relative uncertainty reduction calculated by the change in one sigma
uncertainty used in other MDF studies (e.g. Knorr et al., 2010; Norton et al., 2019), but here the PDFs can be non-normal,
therefore the IQR provides a simple and more representative metric of the uncertainty without the assumption of normality.

### 2.5.2 Model Performance

The model-data fit and predictive skill were evaluated using multiple statistical metrics. For the model output we used the
ensemble median of CARDAMOM sub-samples at each time step. We used Pearson's correlation coefficient (r) to evaluate
to the model skill at capturing the variability, the root mean squared error (RMSE) to evaluate the magnitude of the model-
observed residuals, and mean bias (bias) to evaluate the model prediction bias. The best benchmark for model performance is
whether it can predict data outside of the calibration period, therefore all model skill metrics presented are computed over the
validation period.
The year-to-year variation, or interannual variability (IAV), in carbon cycle processes is better related to climate-carbon
cycle relationships (Piao et al., 2020). Therefore, on top of the monthly variability, we report model skill at capturing IAV in
LAI, NEE, GPP, and $R_e$ over the validation period. We computed the IAV of the annual means, as well as on a seasonal basis.
For the tropical savanna site, AU-How, we define the annual mean by the sites hydrological year which goes from September
to the following August (Hutley and Beringer, 2010), and compute seasonal IAV based on Austral seasons.
Trend analyses were performed using linear regression over the entire simulation period (calibration and validation) to in-
crease temporal coverage. Only months where observations are available were included to ensure a direct comparison between
the modeled and observed trend.

### 2.5.3 Climate Sensitivity

A key aim of this study is to evaluate the climate sensitivity of the carbon cycle following MDF with the two model formu-
lations. We focused on two climatic drivers, temperature and precipitation. Temperature can impact a number of carbon cycle
processes including turnover rates of carbon pools and the physiological response of GPP and LAI. Precipitation impacts plant
available soil water, $W$, and evapotranspiration, $E$. Hence, precipitation can impact GPP in both model formulations via a soil
moisture factor, and Knorr LAI directly via the balance between $W$ and $E$. We also computed the climate sensitivity to vapor
pressure deficit, but this sensitivity was found to be multiple orders of magnitude smaller than temperature and precipitation so
we did not include it in the analysis.
   We used the finite difference method to compute the intrinsic climate sensitivity of LAI and NEE to precipitation and
temperature. All simulations were performed using the forward model, $M$, and CARDAMOM posterior parameter set, $p_{opt}$.
First, the model was run using the prescribed forcing data ($F$) and $p_{opt}$ to generate the control simulation. Second, we perturbed
the precipitation and temperature forcing data (denoted $F'$), independently, over the entire simulation period and ran the model
forward to generate the perturbed simulations. The size of the forcing perturbation needs to be sufficiently small to avoid a
non-linear response in the model. For the precipitation ($P$) perturbation we used $\delta F = \delta P = 1 \times 10^{-8} \ mm \ d^{-1}$ and for the



temperature ($T$) perturbation we use $\delta F = \delta T = 1 \times 10^{-5}\,^{\circ}C$ (applied to both the minimum and maximum air temperature). With the control and perturbed simulations, we computed the derivative of the model output, LAI and NEE, with respect to the climate forcing variables, precipitation and temperature, by:

$$\frac{\delta X}{\delta F} = \frac{M(F', p_{opt}) - M(F, p_{opt})}{\delta F} \tag{15}$$

Where $X$ represents the model output (LAI or NEE) and $F$ represents either precipitation or temperature. This gave a time-series of the intrinsic precipitation and temperature sensitivities of LAI and NEE. We then decomposed these intrinsic sensitivities into (i) seasonal sensitivity by computing the monthly climatology over all simulation years, and (ii) average annual sensitivity by computing the average over the last $n$ years of the simulation period.

To compare and evaluate the relative strength of precipitation sensitivity and temperature sensitivity, it was necessary to normalize the intrinsic sensitivities to a common unit. To do this, we scaled the intrinsic sensitivities by the respective climate variability. For the seasonal sensitivity analysis we multiplied the monthly average intrinsic sensitivity by the monthly interannual variability, computed as the standard deviation of each month in the simulation period. For the annual sensitivity analysis we multiplied the annual average intrinsic sensitivity by the interannual variability, computed as the standard deviation of the 315 annual mean temperature or annual total precipitation. This is calculated using:

$$S_X^F = \frac{\delta X}{\delta F} \times \sigma(F) \tag{16}$$

Where $S_X^F$ provides a measure of the climate sensitivity, $S$, of quantity $X$ to the variability in forcing $F$. This generated four sensitivity metrics, two for LAI ($S_{LAI}^T, S_{LAI}^P$) and two for NEE ($S_{NEE}^T, S_{NEE}^P$), which are evaluated on seasonal and annual time-scales.

## 3 Results and Discussion

### 3.1 Model-Data Fit

The time-series of LAI and NEE at each site is shown in Fig. 2 for the calibration and validation periods, including the two models and the observations. This shows the CARDAMOM posterior PDF for both models at each site. Predictive skill over the validation period (r, RMSE, bias) at each site and for all site data combined is summarized in Fig. 3 and scatter plots in 325 Appendix Fig. A1. Pearson's r shows that NEE temporal variability is better captured by the DALEC$_{\text{Knorr}}$ model at four of the six sites (AU-How, FR-LBr, FR-Pue, GF-Guy), while both models show comparable performance at the remaining two sites (FI-Hyy, US-Ha1), with equal correlations between the two models with all site data combined (r=0.83). The RMSE in NEE for both models is comparable for each site, with smaller residuals from DALEC$_{\text{Knorr}}$ at three sites (AU-How, FR-LBr, FR-Pue), larger residuals at two sites (FI-Hyy and US-Ha1), and equivalent residuals at GF-Guy. For all site data combined, 330 RMSE=0.96 g C m$^{-2}$ d$^{-1}$ for both models. There is a small high bias in model NEE at most sites (<0.5 g C m$^{-2}$ d$^{-1}$), with



DALEC$_{\text{CDEA}}$ showing a slightly lower bias for all site data combined (0.16 g C m$^{-2}$ d$^{-1}$) compared to DALEC$_{\text{Knorr}}$ (0.21 g C m$^{-2}$ d$^{-1}$), indicating the both models slightly underestimate net carbon uptake across sites.

Predictive skill for LAI (Fig. 3) shows that DALEC$_{\text{Knorr}}$ captures a larger proportion of the variability at one site (FR-Pue) and less at three sites (FI-Hyy, FR-LBr, GF-Guy), while both models perform similarly well at the remaining two sites (AU-How, US-Ha1). With all site data combined there is equal correlation with r=0.93. The RMSE for LAI shows that the DALEC$_{\text{CDEA}}$ residuals are smaller than DALEC$_{\text{Knorr}}$ at three sites (AU-How, FI-Hyy, US-Ha1) indicating better performance, while DALEC$_{\text{Knorr}}$ shows better performance at FR-LBr and FR-Pue, and finally both models show similar performance at GF-Guy. Across sites, there is a marginally better performance by DALEC$_{\text{CDEA}}$ (RMSE=0.55 m$^2$ m$^{-2}$) relative to DALEC$_{\text{Knorr}}$ (RMSE=0.57 m$^2$ m$^{-2}$). Both models tend to systematically underestimate LAI, as both models are biased low by 0.23 m$^2$ m$^{-2}$ for all site data.

Both models capture the across site variability in ABGB with r=0.99, as shown in the Appendix Fig. A1. However, DALEC$_{\text{CDEA}}$ has smaller residuals for all site data combined, with a RMSE=472 g C m$^{-2}$ and bias=-217 g C m$^{-2}$, versus a RMSE=952 g C m$^{-2}$ and bias=-609 g C m$^{-2}$ for DALEC$_{\text{Knorr}}$.

Observed and modeled trends for LAI and NEE are shown in Appendix Fig. A3. Observed LAI shows a significant trend at two sites, FR-LBr (p <0.05) and FR-Pue (p <0.001). DALEC$_{\text{CDEA}}$ also shows a significant trend at these two sites but overestimates the magnitude of the trend at FR-LBr by a factor of two and misrepresents the sign of the trend at FR-Pue. DALEC$_{\text{CDEA}}$ also produces a significant positive trend at FI-Hyy (p <0.05) and negative trend at GF-Guy (p <0.001), neither of which are shown by the observations. DALEC$_{\text{Knorr}}$ does not predict a significant trend in LAI at any site. No site shows a significant trend in observed NEE, which the DALEC$_{\text{Knorr}}$ is consistent with. DALEC$_{\text{CDEA}}$, however, predicts a significant positive NEE trend (p <0.05) suggesting a weakening carbon sink, consistent with the strong negative trend in modeled LAI. This suggests that DALEC$_{\text{CDEA}}$ can produce unrealistic trends in both LAI and NEE, whereas DALEC$_{\text{Knorr}}$ tends to be more stable at these time-scales.

Evaluation of the model-data fit to interannual variability (IAV) of LAI and NEE at annual and seasonal time-scales reveal distinct patterns (Appendix Fig. A2). On a seasonal basis, the IAV in LAI for winter, spring, and summer is represented similarly well by both models. However, fall IAV in LAI is captured better by DALEC$_{\text{CDEA}}$. This leads to a slightly better performance by DALEC$_{\text{CDEA}}$ in capturing LAI IAV on an annual basis. For NEE, DALEC$_{\text{Knorr}}$ performs slightly better at capturing IAV across sites with r=0.45 and RMSE=0.33 g C m$^{-2}$ d$^{-1}$, while DALEC$_{\text{CDEA}}$ is r=0.33 and RMSE=0.35 g C m$^{-2}$ d$^{-1}$, with the largest differences in fit occurring during fall.

CARDAMOM is fitting a global cost function which considers all observations in the MDF simultaneously. Therefore, trade-offs can occur between the fit to different observations, both in time and across the observation types (Kato et al., 2013). The results demonstrate that changing the model structure modifies how CARDAMOM converges to an optimal fit for the global cost function. There can therefore be compensatory effects between the fit to LAI and NEE observations. This occurs for the fall IAV in LAI and NEE, where DALEC$_{\text{Knorr}}$ better captures observed fall IAV in NEE, yet it performs worse at capturing observed fall IAV in LAI (Fig. A2). In other cases, a different model structure can lead to improved fit in both LAI and NEE, such as at FR-Pue for DALEC$_{\text{Knorr}}$, suggesting the integrated model structure is overall improved. In any case, assimilating multiple



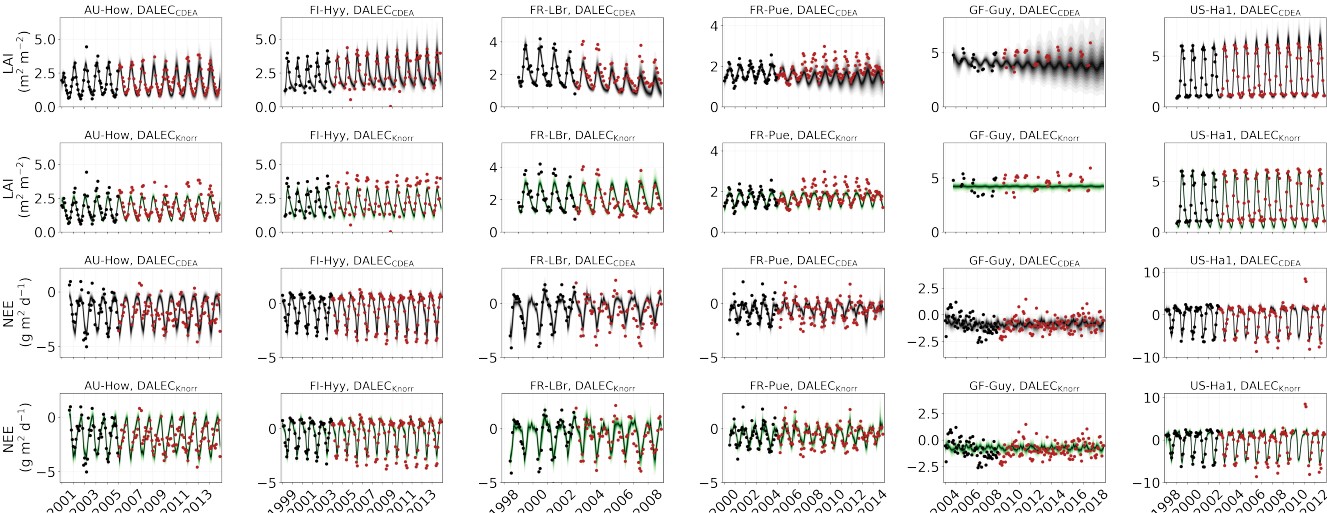

**Figure 2.** Model-data fit shown as time-series at each site and for each CARDAMOM LAI model formulation, for LAI (top panel) and NEE (bottom panel). Assimilated observations are shown as black markers (calibration) and withheld observations as red markers (validation). The gray shading shows the DALEC$_{\text{CDEA}}$ model and the green shading shows the DALEC$_{\text{Knorr}}$ model.

observational data streams simultaneously has benefits over assimilating data streams in separate, sequential steps (Kaminski et al., 2013). A sequential approach requires all uncertainties from each step to be propagated to the next, and this can be challenging when dealing with non-linear models. By assimilating multiple data streams simultaneously, the complementarity of the observations can be exploited and a more consistent view of the terrestrial carbon cycle can be achieved.

## 3.2 Underlying Parameters and Process Constraints

Here, we describe the estimated model parameters and their uncertainty reduction. First, we focus on the Knorr LAI phenology parameters, which govern the link between local climate and phenology, then follow up with a comparison of the remaining shared parameters of the two models.

The parameter uncertainty reduction, calculated as the relative reduction in IQR from the prior to posterior (see methods), for 375 the Knorr LAI phenology parameters differs across sites (Appendix Fig. A5). There are nine process parameters in the Knorr LAI model (excluding initial conditions) across six sites, giving a total of 54 estimated LAI phenology parameters. Of these, 14 parameters show an uncertainty reduction of more than 80%, 15 show an uncertainty reduction between 50-80%, and 8 show an uncertainty reduction between 20-50%. The temperate deciduous forest site, US-Ha1, sees the strongest uncertainty reductions in LAI phenology parameters, with seven of the nine parameters being constrained by more than 50%. The weakest uncertainty 380 reductions in LAI phenology parameters occur at the tropical evergreen forest site, GF-Guy, with just two parameters showing uncertainty reductions greater than 50%.





**Figure 3.** Top panel shows the model-data fit statistics to the assimilated observational data streams, LAI (top) and NEE (bottom), over the validation period. Bottom panel shows the model-data fit statistics to the withheld observational data streams, GPP (top) and $R_e$ (bottom), over the validation period. Markers show the Pearson's correlation coefficient (r), RMSE, and bias, per study site and for all site data combined. The corresponding time-series showing the model-data fits are shown in Fig. 2 for LAI and NEE, and Fig. A4 for GPP and $R_e$. All site data combined scatter plots are shown in the Appendix Fig. A1.



Across all sites, the parameter representing the structural maximum LAI, $\hat{\Lambda}$, is well-constrained, showing an approximate normal posterior PDF and uncertainty reduction of more than 80%, indicating that $\hat{\Lambda}$ is well characterized by the MDF regardless of the site. The parameter for the mean temperature threshold at leaf onset, $T_\phi$, shows strong uncertainty reductions of

66-80% at the four cooler sites (FI-Hyy, FR-LBr, FR-Pue, US-Ha1) and weaker uncertainty reductions of 35-39% at the two warmer tropical sites (AU-How, GF-Guy). This highlights the stronger role of temperature on LAI phenology at the cooler sites. The parameter for the mean photoperiod at leaf senescence, $t_c$, shows strong uncertainty reductions at all sites except the tropical evergreen forest (GF-Guy). There are small reductions in uncertainty of 5-18% for the parameter governing water limitation on LAI (i.e. drought deciduousness), $\tau_W$, with the strongest reductions (15-18%) at AU-How, FI-Hyy, and US-Ha1. The

parameter $\tau_W$ is a proxy for the drought-deciduous behavior of phenology, with larger values indicating a stronger drought-deciduous strategy. The FR-LBr, FR-Pue, and GF-Guy sites show the strongest drought-deciduous behavior, while FI-Hyy and US-Ha1 sites show the weakest. The leaf growth rate parameter, $\xi$, the leaf longevity parameter, $k_{leaf}$, and the background foliar turnover rate, $\theta_{foliar}$, tend to show moderate uncertainty reductions across sites, although GF-Guy shows lower constraint on $\xi$ and $k_{leaf}$.

These results are distinguished from previous MDF studies that used the same phenology model, where phenological types and strategies were differentiated *a priori* (Kaminski et al., 2012; Kato et al., 2013; Knorr et al., 2010). In these studies, each plant functional type used a different prior PDF, while certain parameters were set as constants, effectively switching off some environmental factors that govern the growth triggers and senescence of LAI. As outlined in Knorr et al. (2010), this has advantages if there is sufficient prior knowledge about the species at each study site, however in most cases these

details are based on limited evidence or entirely unknown. Extending these priors beyond well-characterized sites or for global scale analyses with satellite data, requires further assumptions, often by classifying plant functional types which has significant shortcomings (see Van Bodegom et al., 2012). Over confidence in prior knowledge can be problematic, considering the prior PDFs have a significant impact on the inferred parameters and, therefore, the climatic controls of both LAI and NEE. Here, we applied equivalent prior PDFs at all sites, thus allowing CARDAMOM to infer the controls based only on the observations and

local climate. The posterior LAI phenology parameters generally show moderate to strong uncertainty reductions, indicating the ability of CARDAMOM to improve knowledge on phenological controls. The advantage of this approach is that environmental controls on LAI phenology emerge from the MDF system, exemplified by the stronger temperature control of spring leaf onset in cooler climate forests compared to warmer tropical sites. Furthermore, Knorr et al. (2010) fixed *a priori* the LAI water limitation parameter, $\tau_W$, to be zero for cooler forest plant functional types so that LAI is never impacted by water availability.

We find that even for the cooler forest sites, the posterior $\tau_W$ is non-zero, indicating that water limitation plays a role on LAI dynamics, a finding that is also supported by empirical evidence (Buermann et al., 2018; Zhang et al., 2020). We note that switching off the photoperiod regulation process for canopy senescence, as was done in Knorr et al. (2010) for some PFTs, would likely lead to much larger $\tau_W$ values than those in our study, and this would have implications on the link between water availability and LAI dynamics.

There are a number of common parameters between DALEC$_{Knorr}$ and DALEC$_{CDEA}$ (Table A1). The posterior PDFs for these shared parameters are shown in Appendix Fig. A6. The canopy efficiency, a key parameter for GPP, has a higher median in




DALEC$_{Knorr}$ at four of the six sites, while it is the same at FR-Pue and lower at GF-Guy. The inferred carbon-use efficiency for DALEC$_{Knorr}$, equal to the ratio of NPP to GPP (defined by 1-$f_{auto}$), is approximately the same at three sites (FR-LBr, GF-Guy, US-Ha1), higher at two sites (AU-How, FR-Pue) and lower at one site (FI-Hyy). The inferred $LMA$ at each site is
systematically lower in DALEC$_{Knorr}$. Parameters that govern plant carbon allocation and turnover show distinct differences between DALEC$_{CDEA}$ and DALEC$_{Knorr}$. The DALEC$_{Knorr}$ model shows lower fractional allocation of NPP to the foliar pool across sites, consistent with the lower $LMA$ as less carbon is required to maintain the same LAI. DALEC$_{Knorr}$ also shows systematically lower fractional allocation to roots yet a higher fractional allocation to wood. Despite this, the woody residence time is about 60-80% lower in DALEC$_{Knorr}$, as the turnover rate of woody biomass is significantly higher at all sites. Combined,
these differences lead to a lower carbon residence time in vegetation in the DALEC$_{Knorr}$ model.

A key process linked with LAI phenology is litterfall production. We find that the litterfall rates predicted by DALEC$_{CDEA}$ and DALEC$_{Knorr}$ models are substantially different. In DALEC$_{Knorr}$, average annual litterfall rates range between 5.0-12.9 Mg C m$^{-2}$ yr$^{-1}$. These fall well within the range of litterfall rates reported across global forest ecosystems, which range from 1-14 Mg C m$^{-2}$ yr$^{-1}$ (Zhang et al., 2014). The DALEC$_{CDEA}$ model predicts significantly higher litterfall rates at four sites
(AU-How, FI-Hyy, FR-LBr, US-Ha1), ranging between 15.6-37.0 Mg C m$^{-2}$ yr$^{-1}$, which greatly exceeds observed rates documented by Zhang et al. (2014). DALEC$_{Knorr}$ infers litterfall rates at GF-Guy, the tropical evergreen site, that are well within the range observed for that ecosystem type, while DALEC$_{CDEA}$ predicts almost a factor two lower litterfall, which is below the range observed for tropical forests. Overall, this implies that the carbon allocation, turnover, and, subsequently, litterfall is more realistic in DALEC$_{Knorr}$.

Parameters governing the water cycle are generally poorly constrained by the observations and both models tend to infer similar median parameter values. With no hydrology observations used here, there is no expectation of strong constraint on these processes. Despite this, it is notable that DALEC$_{Knorr}$ provides relatively strong constraint (~80%) on the initial water pool size (W$^i$) across sites, perhaps a consequence of the process link between W and the water-limited LAI (Eqs. 4, 5).

### 3.3  Validation of Inferred GPP and R$_e$ Fluxes

The two model structures implemented in CARDAMOM lead to differences in simulated GPP and R$_e$, the component fluxes of NEE. Comparison of the model inferred GPP and R$_e$ to the FLUXNET partitioned GPP and R$_e$ over the validation period is shown in Fig. A4 and summary statistics in Fig. 3. We reiterate that no GPP or R$_e$ data was used during calibration. Almost invariably, DALEC$_{Knorr}$ has a higher correlation, lower RMSE, and lower bias in GPP and R$_e$ relative to DALEC$_{CDEA}$. For GPP, with all site data combined, the model-data fit of DALEC$_{Knorr}$ gives r=0.78, RMSE=2.5 g C m$^{-2}$ d$^{-1}$, bias=-0.8 g C m$^{-2}$ d$^{-1}$,
while DALEC$_{CDEA}$ gives r=0.63, RMSE=3.5 g C m$^{-2}$ d$^{-1}$, and bias=-1.8 g C m$^{-2}$ d$^{-1}$. Qualitatively, the fit to GPP shows similar performance to another recent CARDAMOM study that assimilated GPP and ET data (Smallman and Williams, 2019). The inferred R$_e$ shows a similar RMSE and bias as the GPP model-data fit, although the correlations tend to be lower than for GPP in both models, with r=0.51 for DALEC$_{Knorr}$ and r=0.20 for DALEC$_{CDEA}$. These results indicate that use of DALEC$_{Knorr}$ in CARDAMOM leads to a better representation of the component fluxes underlying NEE, despite a similar fit to assimilated
data streams NEE and LAI.



The improvement in predictive skill of GPP by DALEC$_{\text{Knorr}}$ occurs predominantly at four of the six sites: AU-How, FR-LBr, FR-Pue, and GF-Guy. Both models simulate GPP as a function of the local climate, LAI, the canopy efficiency parameter, and the wilting point parameter that scales with W to impose soil moisture limitation. The difference in inferred GPP between the two models can be traced to the latter three of these factors. We find that the improved prediction of GPP by DALEC$_{\text{Knorr}}$ occurs

due to a higher canopy efficiency at AU-How and FR-LBr, and a weaker soil moisture limitation at FR-Pue and GF-Guy. For the FI-Hyy site, even though the GPP predictions are similar between Knorr and DALEC$_{\text{CDEA}}$, the underlying mechanisms are different as DALEC$_{\text{Knorr}}$ has a 46% higher canopy efficiency, which compensates for the lower LAI. At US-Ha1, the inferred mechanisms controlling GPP are very similar between the two models.

At the two cooler climate forest sites, FI-Hyy and US-Ha1, both DALEC$_{\text{Knorr}}$ and DALEC$_{\text{CDEA}}$ show similar performance

against FLUXNET GPP and R$_e$ data. This suggests that DALEC$_{\text{CDEA}}$ is adequate at inferring NEE component fluxes at these sites. While difficult to trace precisely why this occurs, it may relate to the development of the ACM GPP model which was originally formulated to represent cool climate forests (Williams et al., 1997). It is notable that the model inferred GPP and R$_e$ performs worst at the tropical evergreen site, GF-Guy, with low correlation and the highest residuals (RMSE and bias) of any site. This follows from the relatively poor model-data fit to LAI and NEE at GF-Guy (Fig. 3). At this site the seasonal

variability in LAI and NEE is small relative to the data uncertainty, so seasonal variability carries little weight in the MDF system, potentially making it difficult for CARDAMOM to resolve the seasonal dynamics or its controls. Modeling GPP and LAI at tropical sites has been a longstanding challenge. Evidence has shown that the coupling of leaf age-dependent changes in photosynthetic capacity drives seasonal dynamics GPP, suggesting that models need to separate LAI into cohorts to better represent these leaf demography processes (Wu et al., 2016), a level of complexity that is not currently considered in any

version of the DALEC model.

### 3.4  Climate Sensitivity of LAI and NEE

Here, we present and discuss the inferred climate sensitivities of LAI and NEE for both models. The focus is upon DALEC$_{\text{Knorr}}$ as it provides the relatively more process-based representation of LAI, however, we also explore the results from the DALEC$_{\text{CDEA}}$ model as it provides a useful test case for the effect of model structure on inferred climate sensitivities.

### 3.4.1  Temperature Sensitivities

For both models and all sites, there is a positive $S_{LAI}^{T}$ at an annual timescale, indicating enhanced LAI with an increase in temperature. At all six sites, DALEC$_{\text{CDEA}}$ shows a larger $S_{LAI}^{T}$ than DALEC$_{\text{Knorr}}$, perhaps indicative of the strong dependence of the CDEA formulation on temperature and lack of other direct climate controls. Both the Knorr and DALEC$_{\text{CDEA}}$ models infer the largest annual $S_{LAI}^{T}$ at the two colder forest sites (FI-Hyy, US-Ha1), which is consistent with theoretical understanding of limiting factors on LAI (Caldararu et al., 2014; Richardson et al., 2013). DALEC$_{\text{Knorr}}$ infers a low annual $S_{LAI}^{T}$ at the

remaining four sites, with a median annual $S_{LAI}^{T}$ of near zero for AU-How, FR-Pue, and GF-Guy, while there is a small positive median annual $S_{LAI}^{T}$ at the FR-LBr site. DALEC$_{\text{CDEA}}$, however, infers moderately strong $S_{LAI}^{T}$ at the remaining four





sites, and a median $S_{LAI}^T$ for the two warm tropical sites (AU-How, GF-Guy) that exceed the $S_{LAI}^T$ of two cooler temperate forest sites (FR-LBr, FR-Pue).

The seasonality of the inferred $S_{LAI}^T$ from DALEC$_{\text{CDEA}}$ and Knorr models show distinct differences (Figs. 4-5). DALEC$_{\text{CDEA}}$ tends to show a strong positive $S_{LAI}^T$ year-round, often with a peak during the middle or late part of the growing season. DALEC$_{\text{CDEA}}$ also infers strong positive values during winter, even at the winter-dormant forest sites (FI-Hyy, US-Ha1) which goes against ecophysiological understanding at these sites (Richardson et al., 2013). For DALEC$_{\text{Knorr}}$, the seasonality of $S_{LAI}^T$ is centered on spring onset, with very low $S_{LAI}^T$ at other times of the year. These seasonal patterns of $S_{LAI}^T$ suggest that

DALEC$_{\text{Knorr}}$ is more consistent with empirical understanding of temperature effects on LAI that show the temperature sensitivities is strongest during spring onset (Richardson et al., 2012; Piao et al., 2019).

Generally, at sites with a strong positive $S_{LAI}^T$ at either annual or seasonal timescales, there is also a strong negative $S_{NEE}^T$ (stronger carbon uptake with increased temperature), indicating the impact of LAI climate sensitivity on the sensitivity of NEE and the interrelated nature of these two processes. For DALEC$_{\text{CDEA}}$, the larger positive $S_{LAI}^T$ generally leads to more negative

$S_{NEE}^T$, evident at seasonal timescales and by the negative annual $S_{NEE}^T$ at across all sites. At four sites, the inferred median $S_{NEE}^T$ by the two models differs in sign and magnitude, highlighting the impact of changes to the model formulation of LAI on the inferred temperature sensitivity of NEE. DALEC$_{\text{Knorr}}$ shows more variable annual $S_{NEE}^T$ across sites, as the inferred median annual $S_{NEE}^T$ can be either positive or negative depending upon the site. Seasonally, when there is a strong positive $S_{LAI}^T$ (e.g. spring at FI-Hyy, FR-LBr, and US-Ha1) there is a concomitant negative $S_{NEE}^T$, demonstrating how a positive LAI temperature

sensitivity leads to stronger carbon uptake and that there is a strong seasonal dependence of this link in DALEC$_{\text{Knorr}}$. At these same three sites, other times of the year show a positive $S_{NEE}^T$, suggesting that LAI plays less of a role in in governing $S_{NEE}^T$ outside of spring (Figs. A7-A8).

### 3.4.2 Precipitation Sensitivities

At all sites, $S_{LAI}^P$ in DALEC$_{\text{CDEA}}$ is effectively zero (AU-How, FI-Hyy) or highly uncertain (FR-LBr, FR-Pue, GF-Guy, US-

Ha1). This may reflect the weak process link between water availability and LAI dynamics in DALEC$_{\text{CDEA}}$, considering the connection is mediated via the soil moisture limitation of GPP and subsequent changes in allocation of carbon to the foliar pool, which itself is buffered by the labile carbon pool, making LAI relatively insensitive to changes in water availability. In DALEC$_{\text{Knorr}}$, there is often a weak but consistently positive $S_{LAI}^P$ with a more tightly constrained uncertainty, which implies that increases in precipitation lead to small increases in LAI. This is consistent with the Knorr model formulation for water-

limitation on LAI (Eq. 5), where changes in evapotranspiration ($E$) and/or plant available soil water ($W$) can mediate LAI directly via the $\tau_W$ parameter. More specifically, an increase in the ratio of $W$ to $E$ can increase LAI as there is less water limitation. The opposite is therefore also true, where a decline in the ratio of $W$ to $E$ will lead to a reduction in LAI, for example due to a decline in precipitation. The inferred $S_{LAI}^P$ from DALEC$_{\text{Knorr}}$ is strongest at the AU-How and FR-LBr sites, with smaller $S_{LAI}^P$ values at the three temperate and boreal forest sites, and effectively zero $S_{LAI}^P$ at GF-Guy. Despite the lower

$\tau_W$ inferred at AU-How across sites, there is a relatively stronger $S_{LAI}^P$ which may be due to higher evaporative demand at this tropical site causing larger imbalances between $E$ and $W$.



**Table 2.** Annual temperature sensitivity per site, variable (LAI, NEE), and model in CARDAMOM. The values represent the median annual temperature sensitivity, with the 25th to 75th percentile range in brackets. * LAI sensitivities are scaled by $10^3$.

| | | Annual Temperature Sensitivity per Site | | | | | |
|---|---|---|---|---|---|---|---|
| | | AU-How | FI-Hyy | FR-LBr | FR-Pue | GF-Guy | US-Ha1 |
| $S_{LAI}^T$ * | DALEC$_{CDEA}$ | 17.7 (11.7 to 26.3) | 25.2 (18.1 to 34.6) | 2.9 (1.4 to 5.4) | 5.3 (2.3 to 9.5) | 7.2 (4.2 to 12.7) | 24.1 (19.5 to 29.2) |
| | Knorr model | 0 (0 to 0.3) | 12.4 (1.4 to 25.7) | 0.6 (0.0 to 7.8) | 0 (0 to 1.9) | 0 (0 to 2.6) | 17.4 (1.7 to 49.5) |
| $S_{NEE}^T$ | DALEC$_{CDEA}$ | -8.5 (-11.2 to -6.4) | -3.1 (-4.4 to -1.8) | -0.8 (-1.6 to 0.1) | -0.6 (-1.3 to 0.0) | -0.8 (-1.5 to -0.4) | -0.8 (-2.7 to 2.2) |
| | Knorr model | 0.5 (-1.4 to 3.0) | -1.8 (-5.5 to 0.9) | 1.5 (0.2 to 3.3) | 4.3 (1.7 to 7.3) | -0.1 (-0.6 to 0.7) | 0.8 (-9.4 to 7.3) |

**Table 3.** Annual precipitation sensitivity per site, variable (LAI, NEE), and model in CARDAMOM. The values represent the median annual precipitation sensitivity, with the 25th to 75th percentile range in brackets. * LAI sensitivities are scaled by $10^3$.

| | | Annual Precipitation Sensitivity per Site | | | | | |
|---|---|---|---|---|---|---|---|
| | | AU-How | FI-Hyy | FR-LBr | FR-Pue | GF-Guy | US-Ha1 |
| $S_{LAI}^P$ * | DALEC$_{CDEA}$ | 0.0 (0.0 to 0.0) | 0.0 (0.0 to 50.3) | 58.5 (0.0 to 139.1) | 471.9 (309.5 to 656.5) | 164.7 (0.0 to 373.3) | 0.0 (0.0 to 28.9) |
| | Knorr model | 0.2 (0.0 to 1.8) | 0.1 (0.0 to 0.8) | 0.4 (0.0 to 24.5) | 0.1 (0.0 to 0.7) | 0.0 (0.0 to 0.3) | 0.1 (0.0 to 0.6) |
| $S_{NEE}^P$ | DALEC$_{CDEA}$ | -0.0 (-0.1 to 0) | -1.0 (-5.5 to -0.1) | -30.1 (-53.5 to 0.0) | -96.9 (-146.5 to -68.2) | -20.8 (-36.3 to 0.0) | 2.1 (0.4 to 6.3) |
| | Knorr model | -0.5 (-1.8 to -0.2) | -1.5 (-3.6 to -0.6) | -50.5 (-73.5 to -0.6) | -1.1 (-73.1 to 0.5) | 0.02 (-0.04 to 0.08) | 4.4 (2.0 to 9.8) |

The influence of water availability on Knorr LAI has a clear seasonality at some sites with $S_{LAI}^P$ typically being largest during the mid to late growing season (Figs. 4-5). This seasonal dependence is consistent with recent evidence of late growing season water limitation on productivity at many forest sites (Buermann et al., 2018; Zhang et al., 2020). The evident seasonal

dependence of water limitation on LAI in the Knorr model gives promise for an MDF approach to exploring the compensatory effects of temperature and water limitation on growing season phenology and productivity.

At most sites the inferred annual median $S_{NEE}^P$ from both DALEC$_{CDEA}$ and Knorr models is negative, indicating an increase in net carbon uptake with increased precipitation. Only US-Ha1 shows a consistently positive annual $S_{NEE}^P$, across the uncertainty range and for both models. This is due to the strong positive sensitivity of $R_e$ to precipitation and very small

positive sensitivity of GPP to precipitation inferred at the US-Ha1 site (Fig. A8). The seasonality in $S_{NEE}^P$ shows considerable differences across sites, but it is evidently influenced by $S_{LAI}^P$. For example, DALEC$_{Knorr}$ shows a small positive $S_{LAI}^P$ during the peak growing season at AU-How (Austral summer), and this leads to a stronger sensitivity of summer carbon uptake to precipitation. Similarly, the small positive $S_{LAI}^P$ at FI-Hyy during the late growing season produces a slight seasonal shift in $S_{NEE}^P$ to later in the year. In other sites the low $S_{LAI}^P$ from DALEC$_{Knorr}$ is coupled with a very low $S_{NEE}^P$ (FR-Pue, GF-Guy),

whereas DALEC$_{CDEA}$ infers large and highly uncertain $S_{NEE}^P$ values at these sites. The addition of the process-based Knorr model seems to help stabilize the sensitivity of NEE to precipitation at these two sites, as the exceptionally large and uncertain $S_{LAI}^P$ from DALEC$_{CDEA}$ maps into $S_{NEE}^P$.





**Figure 4.** The seasonal pattern of model-data fit to LAI and NEE, and the inferred climate sensitivity of LAI and NEE to interannual variations in precipitation and air temperature.







**Figure 5.** The seasonal pattern of model-data fit to LAI and NEE, and the inferred climate sensitivity of LAI and NEE to interannual variations in precipitation and air temperature.



### 3.5 Significance and Limitations

Developing models that balance process realism with reliability and robustness is key to better predictions of carbon-climate

feedbacks (Prentice et al., 2015). The climate sensitivity of LAI is an important mechanism of terrestrial carbon-climate feed-backs, as it is closely coupled to both GPP and NEE (Richardson et al., 2013). Here, we implemented a new model for LAI phenology in CARDAMOM that includes processes such as temperature and photoperiod controls on growth and senescence triggers, water limitation on LAI growth rate, and couple LAI dynamics to plant carbon allocation and litterfall. Many previous studies have developed and tested climate-sensitive LAI phenology models against LAI data directly (e.g. Fox et al., 2022;

Jolly and Running, 2004; Stöckli et al., 2008; Viskari et al., 2015). However, in the context of carbon-climate feedbacks it is critical that both model formulation and parameterization of LAI considers its close coupling to other processes in the carbon cycle. By confronting multiple processes with observations simultaneously in CARDAMOM we were able to generate a pa-rameterization that provides a more integrated view of the carbon cycle (Kaminski et al., 2013). Furthermore, by extending the validation beyond the assimilated data streams, we were able to rigorously evaluate model skill. This is a key step toward a

better model representation of the aggregate behavior of the terrestrial carbon cycle (Fisher and Koven, 2020).

Including the two model formulations allowed us to evaluate how the increase in complexity of LAI phenology influenced the fit to assimilated data streams (LAI, NEE, biomass) and wholly withheld data streams (GPP, $R_e$). Evidently, the DALEC$_{CDEA}$ model is a parsimonious approach for fitting data and appears to give reliable predictions of the assimilated data streams (Fig. 3) and even outperforms DALEC$_{Knorr}$ at capturing LAI IAV. The close coupling of LAI with GPP and plant allocation

in DALEC$_{CDEA}$ may provide flexibility in fitting LAI data. However, at four sites DALEC$_{CDEA}$ predicts significant positive (FI-Hyy) or negative (FR-LBr, FR-Pue, GF-Guy) trends in LAI that are not present in the observations, nor are they predicted by DALEC$_{Knorr}$. At one site (FR-LBr), this leads to a significant trend in DALEC$_{CDEA}$ model NEE that is not supported by the observations, suggesting that DALEC$_{CDEA}$ may lead to large biases in the predicted net carbon balance over longer periods. The tighter coupling between climate and LAI phenology in DALEC$_{Knorr}$ may help moderate LAI and GPP dynamics during

the prediction period, preventing unrealistic model trends. This difference in model formulation also leads to a significantly reduced biases in GPP by DALEC$_{Knorr}$, and may imply that the DALEC$_{CDEA}$ model is over-fitting to the LAI data. It is also important to consider that the satellite LAI observations can have systematic biases that we do not consider in the MDF system, as CARDAMOM only considers random errors. Satellite LAI retrievals can be particularly challenging over boreal (e.g. FI-Hyy), tropical (e.g. GF-Guy), and open woody savannas (e.g. AU-How) due to effects such as low solar zenith angle, snow and

cloud contamination, visibility of the understory, and a lack of validation data (Fang et al., 2019). For example, the seasonal amplitude of satellite LAI is often overestimated in boreal evergreen systems (Heiskanen et al., 2012). Other potential error sources include spatial sampling differences between the eddy covariance tower footprint (NEE, GPP, $R_e$), biomass samples, and the footprint of Copernicus satellite LAI data. The limitations of the observations imply caution when over-fitting to any one data stream, and that correcting these sampling biases should allow us to better reconcile model and data. Further

evaluation of the two models suggest that DALEC$_{Knorr}$ better captures temporal variability and magnitude of FLUXNET GPP and $R_e$ data, and predicts more realistic annual litterfall rates compared to a global compilation of observations (Zhang et al.,





2014). Overall, despite the skill of DALEC$_{\text{CDEA}}$ at simulating LAI and NEE over the validation period, when considering the coupling with the carbon balance and the underlying processes, DALEC$_{\text{Knorr}}$ provides more reliable and robust performance.

Future work may explore how additional observations can constrain more uncertain regions of model parameter space. Doing
so will help to further constrain ecosystem carbon cycle dynamics and the sensitivity to climate. In particular, constraining processes governing the water cycle and water availability for plant growth are key future directions of research. Observational constraints on the dynamics of $W$ and $E$ would be beneficial considering the large uncertainty of the parameters associated with these processes (Fig. A6), and could lead to differences in the estimates of $S_{LAI}^P$ and $S_{NEE}^P$. At large scales, observations of terrestrial water storage by the NASA Gravity Recovery and Climate Experiment (GRACE) satellites (Tapley et al., 2004)
can help inform hydrologic parameters and state variables in CARDAMOM (e.g. Massoud et al., 2022). At site scale, in-situ soil moisture measurements and eddy covariance measurements of $E$ may be more useful (Smallman and Williams, 2019). Joint constraint of GPP and LAI may also help to constrain the two underlying controls of GPP, that is light interception which is mediated by LAI and light utilization for photosynthesis which is mediated by plant physiology, and may help further resolve the climate sensitivity of these processes. Satellite observations of solar-induced chlorophyll fluorescence show promise in this
regard (Frankenberg et al., 2011), as they have been shown to inform on model parameters for both GPP and LAI (Norton et al., 2018) and improve spatiotemporal patterns of GPP (Parazoo et al., 2015; Norton et al., 2019). Even without direct observational constraints on GPP, the MDF setup gave reasonable estimates of the underlying fluxes, GPP and R$_{\text{e}}$, with best performance by DALEC$_{\text{Knorr}}$. Separating out NEE into these component fluxes has been a longstanding challenge in carbon cycle science (Schimel and Schneider, 2019). Extending this analysis to more diverse ecosystems is needed to evaluate the
robustness of this result. There are opportunities to extend this to larger scales by using available observational data streams, such as atmospheric inversion estimates of NEE, estimates of biomass from passive microwave and radar sensors, and satellite LAI products (Schimel and Schneider, 2019). Overall, our results demonstrate that when assimilating these data streams into a model, the model process representation significantly affects how information from observations is propagated through to parameters and target processes, and that extending model evaluation to withheld data streams provides critical insight into
model skill.

Joint constraint of LAI and NEE using observations led to important inferences about the climate sensitivity of the ecosystem carbon balance at the six study sites. From DALEC$_{\text{Knorr}}$, most temperate and boreal sites show that $S_{LAI}^T$ occurs almost exclusively during spring onset which leads to a stronger seasonality in $S_{NEE}^T$. At some sites (AU-How, FR-LBr, FR-Pue) the two models infer $S_{NEE}^T$ of opposite sign demonstrating how process representation of LAI leads to differences in optimized
model behavior and response to climate. Furthermore, MDF with DALEC$_{\text{Knorr}}$ infers stronger $S_{LAI}^T$ at colder climate forest sites, as expected from ecophysiological understanding and empirical observations (Richardson et al., 2013), highlighting the ability to infer temperature controls on phenology across biomes using this framework. The influence of water availability on LAI at temperate and boreal sites is strongest at the peak or late in the growing season, whereas at the tropical woodland savanna site (AU-How) it impacts LAI from the beginning to end of the growing season, suggesting biome-specific water
controls on phenology. Further development of data-constrained $S_{LAI}^T$ and $S_{LAI}^P$ will help to reduce the large spread in earth system model predictions of LAI (Mahowald et al., 2016).





The temporal structure of the inferred climate sensitivities has implications for the response of ecosystem carbon cycling to a changing climate. Variability in climate forcing, due to both natural and anthropogenic factors, is not uniform in time or space, with seasonal dependencies on both the variability and trend (Franzke et al., 2020). The inferred sensitivities to temperature and precipitation (Figs. 4-5) show strong seasonality, so the seasonal structure of future climate change will have a strong impact on the response of the carbon cycle. For example, a change in spring temperature forcing will have markedly different impacts on LAI and NEE than an equivalent change in fall temperature forcing due to seasonal differences in the intrinsic sensitivity of the terrestrial carbon cycle. The intrinsic sensitivity to climate is the combination of a number of distinct yet interrelated processes, each of which can have different effects on the net carbon balance. Here, the inferred temperature sensitivities of LAI, $S_{LAI}^T$, are positive definite in all cases. However, the inferred temperature sensitivities of NEE, $S_{NEE}^T$, can be positive or negative depending upon the site and season. The climate sensitivities of NEE integrate over a number of underlying processes, each of which can have differences in sign and magnitude in their sensitivities (e.g. GPP and $R_e$ can respond oppositely to precipitation, Appendix Fig. A7-A8). Constraining these sensitivities using diverse observations provides a robust way toward better representation of carbon-climate feedbacks.

Many phenological processes operate on time scales shorter than the monthly time step used in this analysis. Therefore, accurately resolving specific phenological events such as spring onset or fall senescence events, which can occur within days to weeks, is outside the scope of this study. However, the necessary mechanisms are included in the Knorr LAI model, providing a path toward finer time-scale analyses, which would help to characterize phenological responses to climate and the relationship with NEE. In any case, as outlined by Keenan et al. (2020) data-informed process modeling is key to resolving these processes, as implemented here, so that explicit consideration of processes and observational uncertainties can be mapped onto the inferred climate sensitivities.

## 4 Conclusions

This study demonstrates a holistic approach to model development for the purpose of improving model representation of the climate response of the terrestrial carbon cycle. We integrated a new formulation for LAI phenology into the DALEC terrestrial biosphere model, outlining the coupling to the carbon and water cycles, and performed MDF using CARDAMOM to calibrate the model against diverse Earth observations. Relative to the previous LAI phenology model in DALEC, the new DALEC model showed improved representation of the underlying processes governing the net ecosystem carbon balance, GPP and $R_e$, yet similar performance against the assimilated observations, LAI, NEE, and biomass. This analysis was carried forward to evaluate the data-informed climate sensitivity of the new model structure and parameterization, which showed large changes in the seasonality, sign, and magnitude of LAI and NEE sensitivities to temperature and precipitation. The added process realism of LAI phenology in DALEC/CARDAMOM provided more realistic and robust predictions of the terrestrial carbon cycle and its response to climate, highlighting the important role that LAI phenology plays in representing the terrestrial carbon cycle especially when considered in a MDF system.





*Code and data availability.* Model code for DALEC/CARDAMOM and the data used in this study is available upon request.

# Appendix A: DALEC Model Parameters

**Table A1.** DALEC model parameters that are shared, with the same physical meaning in both DALEC$_{CDEA}$ and DALEC$_{Knorr}$. This includes process parameters and initial conditions along with their prior range. Note: W = plant-available soil water pool, W$_u$ = plant-unavailable soil water pool. The allocation fraction to wood $f_{wood}$ is computed as 1 - $f_{auto}$ - $f_{lab}$ $f_{fol}$ - $f_{root}$ in DALEC$_{CDEA}$, and 1 - $f_{auto}$ - $f_{lab}$ - $f_{root}$ in DALEC$_{Knorr}$.

| Class | # | Description | Symbol | Prior Range |
|---|---|---|---|---|
| Vegetation carbon | 1 | Canopy GPP efficiency | $c_{eff}$ | 5-50 |
| | 2 | Wilting point | $\omega$ | 1-10000 |
| | 3 | NPP fraction to $C_{lab}$ [a] | $f_{lab}$ | 0.01-0.5 |
| | 4 | Leaf carbon mass per area | $LMA$ | 5-200 |
| | 5 | Fraction of GPP respired | $f_{auto}$ | 0.2-0.8 |
| | 6 | NPP fraction to $C_{root}$ [a] | $f_{root}$ | 0.01-1 |
| | 7 | Wood carbon turnover rate | $\theta_{wood}$ | 0.000025-0.001 |
| | 8 | Root carbon turnover rate | $\theta_{root}$ | 0.0001-0.01 |
| Litter & soil carbon | 9 | Litter carbon turnover rate | $\theta_{lit}$ | 0.0001-0.01 |
| | 10 | Soil carbon turnover rate | $\theta_{soil}$ | 0.0000001-0.001 |
| | 11 | Turnover rate for litter-soil transfer | $\theta_{lit2soil}$ | 0.0001-0.01 |
| | 12 | Decomposition temp. rate | $\theta$ | 0.018-0.08 |
| | 13 | Moisture factor for decomposition | $s_p$ | 0.01-1 |
| Water cycle | 14 | Underlying water-use efficiency | uWUE | 0.5-30 |
| | 15 | W runoff focal point | $W^{Qmax}$ | 1-100000 |
| | 16 | W$_u$ runoff focal point | $W_u^{Qmax}$ | 1-100000 |
| | 17 | W to W$_u$ runoff fraction | $h2o_{xfer}$ | 0.01-1 |
| Initial conditions | 18 | Initial $C_{lab}$ | $C_{lab}^i$ | 1-2000 |
| | 19 | Initial $C_{fol}$ | $C_{fol}^i$ | 1-2000 |
| | 20 | Initial $C_{root}$ | $C_{root}^i$ | 1-2000 |
| | 21 | Initial $C_{wood}$ | $C_{wood}^i$ | 1-100000 |
| | 22 | Initial $C_{lit}$ | $C_{lit}^i$ | 1-2000 |
| | 23 | Initial $C_{soil}$ | $C_{soil}^i$ | 1-200000 |
| | 24 | Initial W pool | $W^i$ | 1-10000 |
| | 25 | Initial W$_u$ pool | $W_u^i$ | 1-10000 |





## Appendix B: CDEA Model Description and Parameters

The leaf onset factor ($\phi_{onset}$), which is used to compute the carbon pool transfer from $C_{lab}$ to $C_{fol}$, was originally defined in Bloom and Williams (Eq. A7, 2015) but was subsequently updated to include a variable residence time for $C_{lab}$ (Bloom et al., 2020), with a formulation of


$$\phi_{onset}(t) = \frac{2}{\sqrt{\pi}} \cdot \left( \frac{ln(\theta_{lab}) - ln(\theta_{lab} - 1)}{c_{onset}\sqrt{2}} \right) \cdot e^{-(sin(\frac{t - d_{onset} + osl}{s}) \cdot \frac{2s}{c_{ronset}\sqrt{2}})^2} \tag{B1}$$

where $\theta_{lab}$, $c_{onset}$, and $d_{onset}$ are time-invariant parameters (Table B1), and $osl$ is calculated by

$$osl = offset(\theta_{lab}, \frac{c_{ronset}\sqrt{2}}{2}) \tag{B2}$$

and $s = \frac{365.25}{\pi}$.

The leaf fall factor ($\phi_{fall}$), which is used to compute the carbon pool transfer from $C_{fol}$ to $C_{lit}$ (i.e. litter production), was
originally defined in Bloom and Williams (Eq. A8, 2015) but was subsequently updated to include a variable residence time for $C_{fol}$ (Bloom et al., 2020), with a formulation of:

$$\phi_{fall}(t) = \frac{2}{\sqrt{\pi}} \cdot \left( \frac{ln(\theta_{fol}) - ln(\theta_{fol} - 1)}{c_{rfall}\sqrt{2}} \right) \cdot e^{-(sin(\frac{t - d_{rfall} + osf}{s}) \cdot \frac{2s}{c_{rfall}\sqrt{2}})^2} \tag{B3}$$

where $\theta_{fol}$, $c_{rfall}$, and $d_{rfall}$ are time-invariant parameters (Table B1), and where $osl$ is calculated by

$$osf = offset(\theta_{fol}, \frac{c_{rfall}\sqrt{2}}{2}) \tag{B4}$$

The $C_{fol}$ is updated at each time step by:

$$C_{fol}(t+1) = f_{fol}NPP(t) + \phi_{onset}(t)C_{lab}(t) + (1 - \phi_{fall}(t))C_{fol}(t) \tag{B5}$$

And $C_{lab}$ is updated at each time step by:

$$C_{lab}(t+1) = f_{lab}(NPP(t) - f_{fol}NPP(t)) + (1 - \phi_{onset}(t))C_{lab}(t) \tag{B6}$$





**Table B1.** DALEC$_{CDEA}$ LAI phenology parameters including process parameters and initial conditions, along with their prior range. $^a$ This parameter is used in DALEC$_{CDEA}$ to allow for direct carbon allocation to $C_{fol}$, whereas DALEC$_{Knorr}$ does not, as $C_{fol}$ is only supplied with carbon from $C_{lab}$. $^b$ Uses a circular prior range that extends beyonda 365.25 to prevent edge jumping during optimization (e.g. Dec 31 to Jan 1), as this parameter is used in a sine function with an annual period, so the actual day-of-year value can be computed as modulo 365.25.

| Class | # | Description | Symbol | Prior Range | Units |
|---|---|---|---|---|---|
| | 1 | NPP fraction to $C_{fol}$ $^a$ | $f_{foliar}$ | 0.01-0.5 | - |
| | 2 | $C_{lab}$ (leaf) lifespan | $\theta_{fol}$ | 1.01-8 | - |
| | 3 | $C_{lab}$ lifespan | $\theta_{lab}$ | 1.01-8 | - |
| | 4 | Peak day of year for $C_{lab}$ turnover | $d_{onset}$ | 365.25-1461$^b$ | days |
| LAI phenology | 5 | Peak day of year for $C_{fol}$ turnover | $d_{rfall}$ | 365.25-1461$^b$ | days |
| | 6 | $C_{lab}$ release period | $c_{ronset}$ | 30.4375-100 | days |
| | 7 | Leaf fall period | $c_{rfall}$ | 30.4375-150 | days |
| | 8 | Leaf carbon mass per area | $LMA$ | 5-200 | g C m$^{-2}$ |
| | 9 | Initial $C_{lab}$ | $C_{lab}^i$ | 1-2000 | g C m$^{-2}$ |
| | 10 | Initial $C_{fol}$ | $C_{fol}^i$ | 1-2000 | g C m$^{-2}$ |

## Appendix C: Knorr Model Description

The labile production flux is computed by:

$$F_{labprod}(t) = (GPP - R_a)f_{lab} = NPP f_{lab} \tag{C1}$$

where $GPP$ is the gross primary productivity, $R_a$ is the autotrophic respiration, $NPP$ is the net primary productivity, and $f_{lab}$ is a parameter representing the fraction $NPP$ allocated to the labile pool.

The quadratic smoothing function is given in Knorr et al. (Eq. 25, 2010), we use $\eta$=0.99.


$$\upsilon(x,y) = \frac{x + y - \sqrt{(x+y)^2 - 4\eta xy}}{2\eta} \tag{C2}$$



**Table C1.** DALEC$_{\text{Knorr}}$ LAI phenology parameters including process parameters and initial conditions, along with their prior range. $^a$ Only during canopy senescence. $^b$ Defined as the fraction of $LAI_{max}$.

| Class | # | Description | Symbol | Prior Range | Units |
|---|---|---|---|---|---|
| LAI phenology | 1 | Mean temperature at leaf onset | $T_\phi$ | 268.15-323.15 | °K |
| | 2 | Spatial range of $T_\phi$ | $T_r$ | 0.1-10 | °K |
| | 3 | Linear growth constant | $\xi$ | 0.001-0.5 | days$^{-1}$ |
| | 4 | Inverse of leaf longevity$^a$ | $k_{leaf}$ | 0.001-0.5 | days$^{-1}$ |
| | 5 | Max intrinsic LAI | $LAI_{max}$ | 0.1-10 | m$^2$ m$^{-2}$ |
| | 6 | Length of dry spell before leaf shedding | $\tau_W$ | 0.1-300 | days |
| | 7 | Mean day length at leaf shedding | $t_c$ | 2-22 | hrs |
| | 8 | Spatial range of $t_c$ | $t_r$ | 0.1-6 | hrs |
| | 9 | Background leaf turnover rate | $\theta_{foliar}$ | 0.001-0.1 | - |
| | 10 | Leaf carbon mass per area | $LMA$ | 5-200 | g C m$^{-2}$ |
| | 11 | Initial $C_{lab}$ | $C_{lab}^i$ | 1-2000 | g C m$^{-2}$ |
| | 12 | Initial $C_{fol}$ | $C_{fol}^i$ | 1-2000 | g C m$^{-2}$ |
| | 13 | Initial air temp. memory | $T^i$ | 268.15-323-15 | °K |
| | 14 | Initial $LAI_W$ $^b$ | $LAI_W^i$ | 0.01-1 | - |




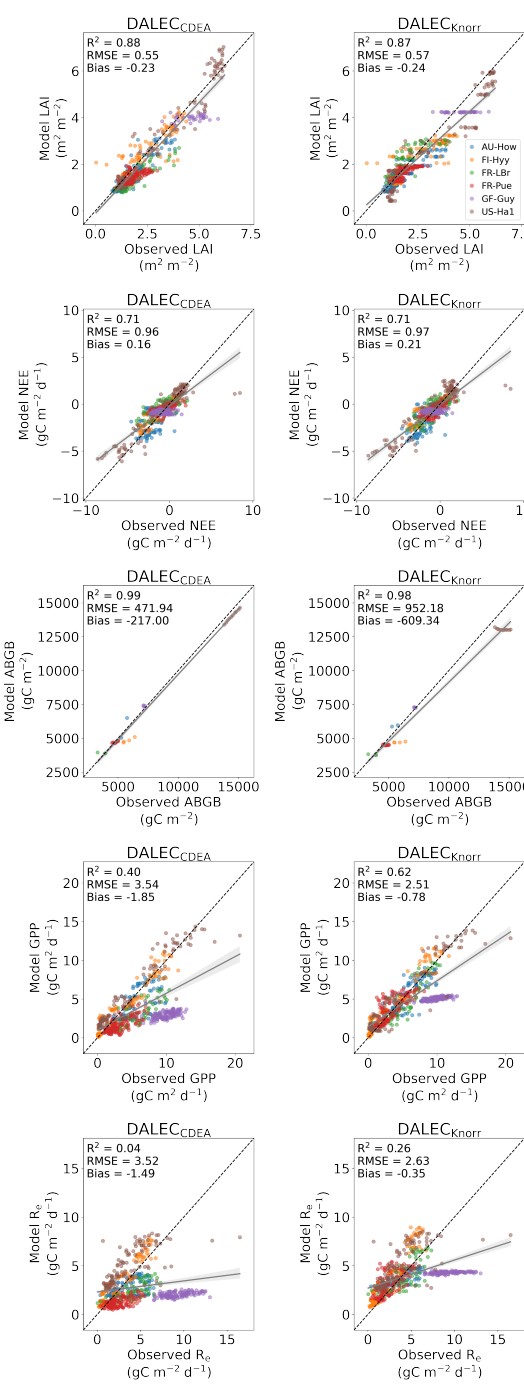

**Figure A1.** Model-data fit to assimilated observations (NEE, LAI, ABGB) and wholly withheld observations (GPP, $R_e$) over the validation period, for the DALEC$_{CDEA}$ model (left) and DALEC$_{Knorr}$ model (right). Each color represents a different site.







**Figure A2.** Model-data fit statistics for the interannual variability on an annual and seasonal basis against the assimilated observations (LAI, NEE) and wholly withheld data (GPP, $R_e$) over the validation period, with all sites combined. Markers show the Pearson's correlation coefficient (r), RMSE, and bias, per study site and for all site data combined.





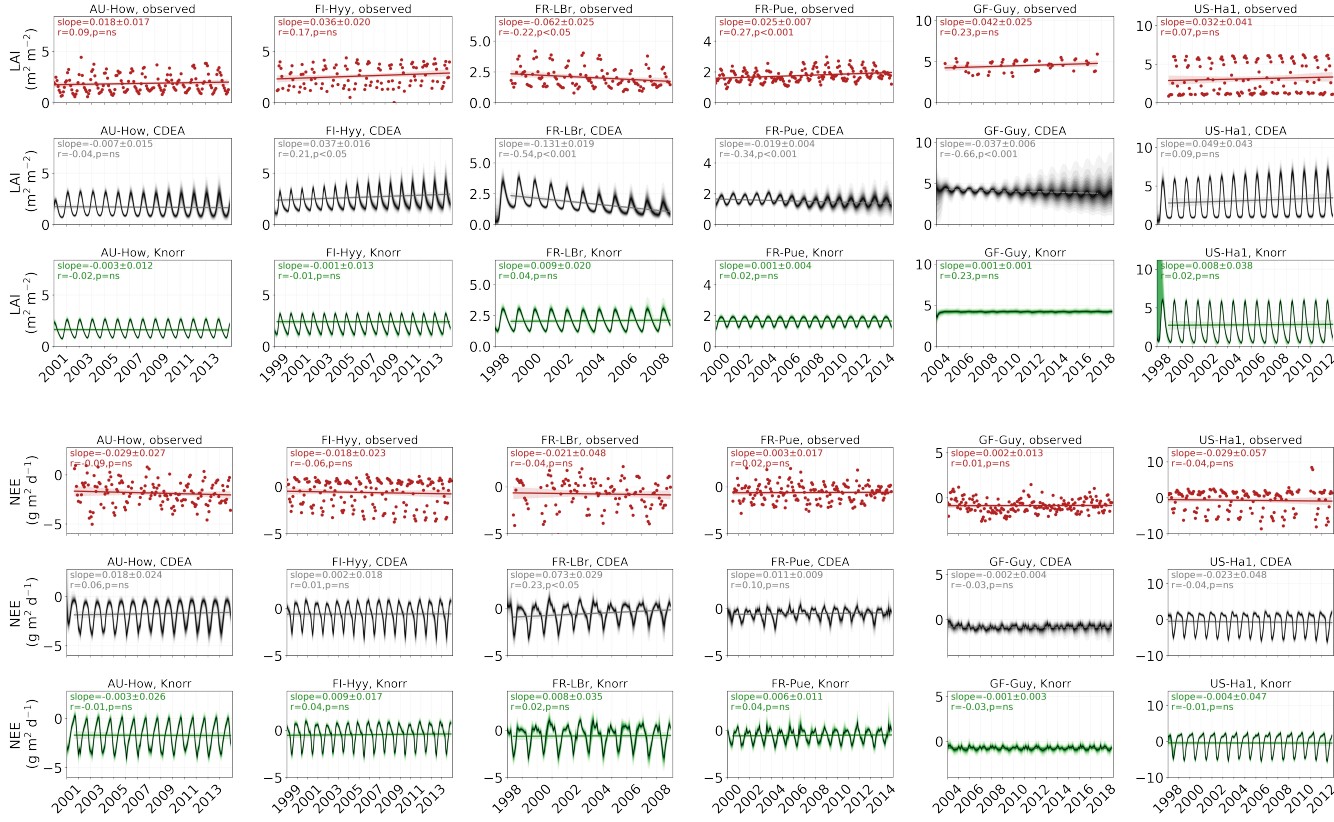

**Figure A3.** Observed and modeled trends over the calibration and validation period for LAI and NEE, including the observations (red dots), DALEC$_{CDEA}$ model (grey line and shading), and DALEC$_{Knorr}$ model (green line and shading). The slope, Pearson's r, and significance level for the linear regression is shown in each panel of the figure.

*Author contributions.* AJN and AAB designed the research. AJN conducted the modeling and analysis, with support from AAB, PAL, and SM. LTS processed the observational data. AJN prepared the manuscript and handled revisions. All authors contributed to interpretation and manuscript revisions.



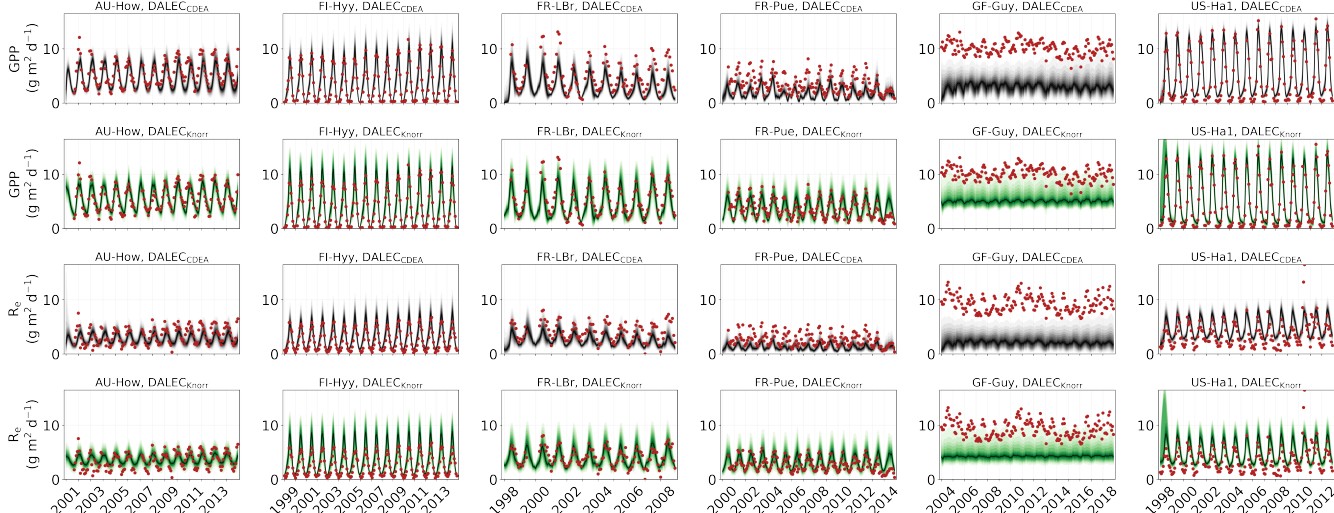

**Figure A4.** Model-data fit shown as time-series at each site and for each CARDAMOM LAI model formulation, for GPP (top panel) and $R_e$ (bottom panel). Both GPP and $R_e$ data were withheld from the MDF for validation. Observations are shown in red markers (validation data). The gray shading shows the DALEC$_{CDEA}$ model and the green shading shows the DALEC$_{Knorr}$ model.

*Acknowledgements.* A portion of this research was supported by the Jet Propulsion Laboratory, California Institute of Technology, under a contract with the National Aeronautics and Space Administration. Part of the funding for this study was provided through a NASA Carbon Cycle Science grant (#NNH20ZDA001N-CARBON). Alexander Norton, Anthony Bloom, Nicholas Parazoo, Paul Levine, Shuang Ma, and Renato Braghiere were all supported by the Jet Propulsion Laboratory, California Institute of Technology. NCP acknowledges funds and support provided under the NASA Earth Science Division (ESD) MEASURES and Arctic Boreal Vulnerability Experiment (ABoVE) programs, and the NSF Arctic Natural Sciences program. TLS was supported by the UK's National Centre for Earth Observation.





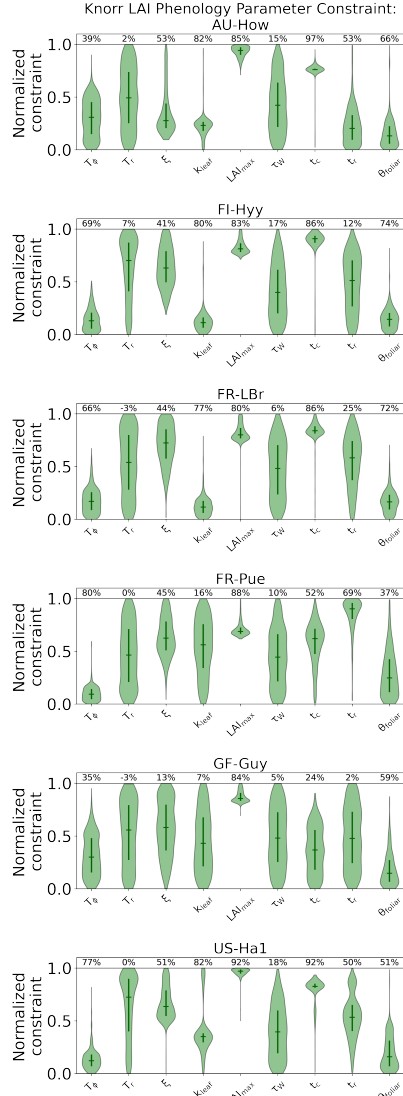

**Figure A5.** LAI phenology posterior parameter PDFs for the Knorr model, shown as the log-normalized PDFs between the minimum (normalized constraint=0) and maximum (normalized constraint=1) parameter bounds. The violin plot shows the full posterior PDF, the solid vertical bars indicate the 25th to 75th percentile range (IQR), and the horizontal solid lines indicate the median. The percentages at the top of the figure indicate the parameter uncertainty reduction from prior to posterior, reported as the reduction in log-normalized IQR (see methods).



**Figure A6.** Posterior parameter PDFs for the common process parameters of the DALEC$_{CDEA}$ model (gray) and DALEC$_{Knorr}$ model (green), excluding initial conditions. Violin plots of 'normalized contraint' show the log-normalized PDFs between the minimum (normalized constraint=0) and maximum (normalized constraint=1) parameter bounds. The violin plot shows the full posterior PDF, the solid vertical bars indicate the 25th to 75th percentile range (IQR), and the horizontal solid lines indicate the median. The percentages at the top of the figure indicate the parameter uncertainty reduction from prior to posterior, reported as the reduction in log-normalized IQR (see methods).





**Figure A7.** The seasonal pattern of model simulated GPP and $R_e$, and the inferred climate sensitivity of GPP and $R_e$ to interannual variations in precipitation and air temperature.





**Figure A8.** The seasonal pattern of model simulated GPP and $R_e$, and the inferred climate sensitivity of GPP and $R_e$ to interannual variations in precipitation and air temperature.



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
