# Peer review of "Improved process representation of leaf phenology significantly shifts climate sensitivity of ecosystem carbon balance"

_EGUsphere, 2022_

## Author Response (AR1)

**Author response for:** Norton et al., Improved process representation of leaf phenology significantly shifts climate sensitivity of ecosystem carbon balance, egusphere-2022-1265

We would like to sincerely thank the reviewers for their positive and constructive feedback. Please see below for our responses (in blue text) to each comment and details on how we integrated these comments into the revised manuscript.

**Reviewer Comment 1 (RC1)**

General comments:

This manuscript seeks to evaluate the influence of different representations of leaf phenology on modeled terrestrial carbon cycle estimates. The manuscript compares two LAI phenology formulations---one with no climate controls (CDEA, the default in DALEC), and one where timing and growth are influenced by climate (Knorr et al. 2010, with some DALEC-specific modifications). This manuscript uses the CARDAMOM terrestrial ecosystem modeling and data assimilation framework, calibrated jointly against LAI (Copernicus EO 1km product) and NEE (FLUXNET2015) measurements and validated against tower-based GPP and RE (FLUXNET, based on night-time partitioning) and in-situ biomass measurements (with site-specific allometric scaling). The analysis is performed at 6 FLUXNET sites spanning a variety of biomes. Results show that the climate-driven phenology scheme improved predictions of GPP, RE, and litterfall. The climate-driven phenology scheme also led to different NEE sensitivity to precipitation and temperature.

Overall, I found this to be a solid, well-executed study. The science topic --- representations of LAI phenology in vegetation models --- is important and relevant. The modeling approach, and the methods for calibration, validation, and sensitivity analysis, are well-explained and sound. The results are compelling and well-interpreted and contextualized in the literature. I have a few minor comments related to presentation (see detailed comments below), but I think the overall quality of this study is good.

We thank the reviewer for their thoughtful comments and encouraging feedback. We have gone through their specific comments below and addressed them point by point, while noting where we have made changes to the manuscript.

Detailed comments:

[Line 7, "biomass"]

Based on the methods, I think the model is *validated* against biomass but only calibrated against LAI and NEE (i.e., only LAI and NEE appear in the likelihood).

Thanks for pointing this out. You are correct that only the LAI and NEE appear in the likelihood. That is an error on our part. The biomass should also appear in the likelihood as it is included in

the calibration step, as described in lines 103-112. We have corrected the description of the likelihood in the methods.

[Line 40]

Somewhere in here, you might also consider citing Wheeler & Dietze 2021 (DOI: 10.5194/bg-18-1971-2021).

Thanks for pointing out this interesting study. We have incorporated it into the introduction.

[Line 56, "Bayesian data assimilation"]

Although technically not inaccurate, I find the terms "data assimilation" and "Model data fusion" to be somewhat vague and potentially misleading in this context. Here and elsewhere, I suggest more precise terminology such as (Bayesian) "calibration", "optimization", or "parameter data assimilation", to distinguish what is done here (tuning of model *parameters* that affect the entire course of the simulation) from *state* data assimilation (a stepwise process in which model *states* at a particular time and place are tuned to better match observations, e.g., via Kalman filter, as is done in reanalysis products). (Admittedly, Macbean et al. 2016 and many others also use "data assimilation" this way, so this is not a problem unique to this study.)

We agree that the terminology in this field can be convoluted, especially to newcomers. Your suggestion is taken onboard and we have modified the paragraph to state "we use a Bayesian parameter data assimilation system…" for specificity.

[Equations 3-5, 10, others]

You might consider using explicit multiplication symbols (x or dot), spacing, fonts (e.g., non-italic font for symbols like LAI), different brackets (e.g., hard brackets for indexes), or different kinds of symbols (e.g., Greek vs. Latin, capital vs. lowercase) to more clearly distinguish between multiplication, function calls, indexing, and multi-letter acronyms (e.g., in equation 2, Phi refers to the Normal CDF called on the fraction in parentheses, whereas in equation 3, the lowercase chi is presumably multiplied by the LAI difference; WLAI isn't immediately obvious as W x LAI).

Very good suggestion. This should help with clarity of the equations. We have implemented all of the reviewers suggestions to improve readability of the mathematics.

[Equation 7]

This probably needs the (t) index for the terms on the right?

Yes, good catch, thank you. Corrected in the new version.

[Equation 8]

C(lab) here probably needs a time index (t-1?)

Yes, that is correct. Thanks for pointing that out. Corrected in the new version.

[Line 476, "positive ST_LAI"]

This is slightly misleading, since the Knorr formulation predicts near-zero ST_LAI in the warmer sites (which is what one would hope!). I suggest tweaking this sentence to highlight the differences across formulations and sites.

Yes, I can understand how that sentence can be misleading and appear to conflict with the results. The point we are trying to make is that, for none of the posterior samples is there a negative ST_LAI. We have rephrased this to say "The median $S_{LAI}^T$ on an annual timescale ranges from zero to strongly positive depending upon the model and site". Furthermore, we have added this sentence to the end of the paragraph "In neither model does the LAI show a negative sensitivity to temperature" to highlight the point that LAI temperature sensitivities can only ever be positive from these two models.

[Line 634, "available upon request"]

EGUsphere doesn't have a data use policy (or at least, I couldn't find one) so this is technically not a violation. However, I personally feel that "availability upon request" is unacceptable data sharing policy for modern scientific publications. Unless there is a clear and compelling reason (e.g., government mandate, conservation risk, etc.; if there is such a limitation, it should be explicitly stated), data should to be deposited in a publicly available repository such as Dryad, FigShare, or Open Science Framework. The importance and benefits of open data have been widely documented over the last decade; among the most recent examples is Noy and Noy 2019 (https://doi.org/10.1038/s41563-019-0539-5), and journals are increasingly requiring code and data sharing as a precondition for publication (e.g., AGU data policy -- https://www.agu.org/Publish-with-AGU/Publish/Author-Resources/Data-and-Software-for-Authors; GMD data policy -- https://www.geoscientific-model-development.net/policies/code_and_data_policy.html).

Yes, we agree that open data sharing should be the norm. To address this, we have created (i) public, read-only release of the CARDAMOM model code, and (ii) a citable repository for the associated model input/driver data, model outputs, and post-processing analysis code to generate the figures. Please see the section Code and Data Availability in the revised manuscript for the details.

**Reviewer Comment 2 (RC2)**

Generally this paper is a good discussion on evaluation of the inferred climate sensitivity of LAI and NEE with the models, added complexity shifts the sign, magnitude, and seasonality of NEE sensitivity to precipitation and temperature. The research showed the benefit of process complexity when inferring underlying processes from Earth observations and in representing the climate response of the terrestrial carbon cycle.

Thank you for the encouraging comments and recognition of the benefit of this research.

Page 6, why the LAI Phenology Models cannot be used only one model.

Based on this reviewer's comment, we believe there is some confusion around the implementation of the "model" and the data-fusion system. To clarify, both LAI phenology models are implemented as modules (or "components") within the same ecosystem model (DALEC). Each version of DALEC (DALEC_Knorr and DALEC_CDEA) is then used within the CARDAMOM model-data fusion system, which tunes parameters and initial conditions of DALEC to optimally fit the prior information and new observational information. We hope this clarifies any issues for the reviewer.

Page 14, Figure 2: you can highlight which graphs are the underlining ones and you've discussed.

We are unsure what the reviewer is requesting. All of the graphs are important to present as they represent the sites studied and model-observed comparisons. Each graph is also discussed in the main text, and we refer to the study site codes which are indicated in the title of each plot of Figure 2. We hope this clarifies the reviewer's concern.

Page 15, Figure 3: the RMSE units should be labeled clear.

The units for RMSE in Figure 3 are clearly labeled in the same way as the bias metrics and align with the units used in the other figures. We believe this figure fulfills the requirements in Biogeosciences guidelines (https://www.biogeosciences.net/submission.html#figurestables). However, we will happily make modifications as per the editors request.

---

## Author Response (AR2)

**Author response for:** Norton et al., Improved process representation of leaf phenology significantly shifts climate sensitivity of ecosystem carbon balance, egusphere-2022-1265

**To the editor**:

Many thanks for your feedback. We are pleased with the streamlined review process. Below, I've outlined the changes you have requested to fully address Reviewer 2's comments. This includes minor changes in the Introduction and Methods sections to distinguish the modeling and model-data fusion components of the study i.e. the LAI models (Knorr, CDEA), the terrestrial biosphere model (TBM i.e. DALEC) and the model-data fusion system (CARDAMOM)..

1. In the Introduction and Methods sections, any reference to the LAI model is changed to "LAI submodel". For example, at the end of the Introduction the text now clearly states "We implement a prognostic, climate-sensitive LAI **submodel** into a TBM and benchmark this against an empirical diagnostic LAI **submodel** used in a previous version of the same TBM".
2. In the Methods section 2.2 Model-data fusion, we have modified the text to make the distinction between components clear: "CARDAMOM is a Bayesian MDF system, used to retrieve time-invariant parameters and initial conditions for the Data Assimilation Linked ECosystem (DALEC) TBM".
3. Furthermore, we have added some text under the section Model Description to state: "we describe the two separate implementations of LAI phenology used in this study that are linked to same representation of carbon and water cycles **i.e. the same TBM but different LAI phenology submodels**".
4. Finally, under the section Model Description and subsection Knorr Model, we have removed the reference to "CARDAMOM" and replaced it with "DALEC", as that is the TBM we're referring to.